# Apoptotic bodies in phytoplankton suggest evolutionary conservation of cell death mechanisms

Luisa Corredor [1,4,5] ✉, Georgia Antonia Vergou [1,5], Vladimír Skalický [2], Ioanna Antoniadi [2], Benjamin J. Wheaton [3], Karin Ljung [2], András Gorzsás [1] & Christiane Funk [1] ✉

Programmed Cell Death (PCD) in eukaryotes is a regulated process occurring during development, cell differentiation and aging. Apoptosis is a particularly well studied morphotype of PCD, only observed in animal cells (metazoan). Its most definitive hallmark is the formation and release of membrane-enclosed extracellular vesicles called Apoptotic Bodies (ABs). Although apoptotic-like features have been described in plants, yeast, protozoa and phytoplankton, the production of ABs has been thought to be limited to multicellular animals. Here we report the production and release of extracellular ABs in a non-metazoan unicellular eukaryote, the cryptophyte alga *Guillardia theta*. Morphologies of *G. theta* cells during aging and pharmacologically-induced cell death confirm the presence of ABs and apoptosis in phytoplankton. *G. theta* ABs have similar composition to metazoan ABs, carrying DNA, proteins, lipids, carbohydrates, fragments of organelles and cytosol portions. Our results demonstrate that *G. theta*, a microalga that arose from secondary endosymbiosis, experiences apoptotic cell death in physiological conditions, similar to animal cells. Since secondary endosymbiosis occurred prior to the origin of multicellularity, our discovery questions the evolutionary origin of PCD.

Death is the one fate shared by all living cells and is a certainty for all living organisms. In eukaryotes, cell death is part of normal development, homeostasis and pathogenic processes[1]. It is commonly separated into broader categories of regulated cell death such as apoptosis, necroptosis, parapoptosis and lysosome- and autophagy-mediated cell death[2, 3]. Apoptosis, a particularly well studied morphotype of programmed cell death (PCD) in metazoans, occurs during normal physiological processes such as development, cell differentiation and aging[2–4]. In the dying cell, a family of cysteine-proteases called caspases orchestrates apoptotic signaling triggering several morphological and biochemical changes[5]. In later stages of apoptosis, cells form membrane-enclosed vesicles called apoptotic bodies (ABs)[3,4], which are considered to be the most definitive hallmark of apoptosis[6,7]. Among multicellular organisms only animal cells produce ABs, whereas plants, despite exhibiting apoptotic-like features[8,9], do not. Instead, plant cells remain encapsulated by rigid cell walls with no evidence of apoptotic body production[8]. In unicellular eukaryotes (e.g., phytoplankton and yeast), the need for a genetically encoded process leading to cellular self-destruction is not obvious, yet PCD has been found to enhance genetic and population fitness[10]. During aging

[1]Department of Chemistry, Umeå University, Umeå, Sweden. [2]Umeå Plant Science Centre (UPSC), Department of Forest Genetics and Plant Physiology, Swedish University of Agricultural Sciences, Umeå, Sweden. [3]Department of Medical Translational Biology, Umeå University, Umeå, Sweden. [4]Present address: Department of Food, Environmental and Nutritional Sciences, University of Milan, Milan, Italy. [5]These authors contributed equally: Luisa Corredor, Georgia Antonia Vergou. ✉e-mail: luisa.corredor@unimi.it; christiane.funk@umu.se

or development of cultures and colonies, single cells are part of a multicellular population and the altruistic death of old and damaged cells represents an adaptation to benefit the population[10,11]. Dead cells can release limiting nutrients, differentiation molecules and unidentified pro-survival factors that can be metabolized, promoting the survival of younger and fitter cells[10]. In unicellular organisms, enzymes called metacaspases, structural homologs of caspases, have been linked to PCD. Some features of PCD have been shown in yeast[2,12], protozoa[13] and phytoplankton such as chlorophytes[14–17], diatoms[18,19], dinoflagellates[20] and even cyanobacteria[21,22]; however, no evidence of apoptotic body production has previously been found.

Cryptophytes, unicellular microalgae that evolved from secondary endosymbiosis, have the outstanding ability to adapt to oligotrophic environments and thrive in limited light conditions. They are key players in the aquatic food chain, primary producers in both freshwater and marine habitats[23,24] and, being phytoplankton, they contribute to nearly 50% of the global annual carbon-based primary productivity[25]. The cryptophyte alga *Guillardia theta* (*G. theta*) has emerged as a model organism within this group[23]. We investigated the hallmarks of PCD during standard growth conditions in *G. theta* and − for the first time in a photosynthetic organism − observed the production of apoptotic bodies. Having performed a detailed characterization of *G. theta* cells and ABs (Gt-ABs) by confocal microscopy, transmission electron microscopy (TEM), Fluorescence Activated Cell Sorting (FACS), confocal Raman Spectroscopy (RS), pharmacological and molecular studies, we show the first definitive evidence of apoptosis in unicellular eukaryotes and the production of apoptotic bodies. PCD might be a conserved process in unicellular organisms, which has been retained through secondary endosymbiosis.

## Results

### Apoptosis and the production of apoptotic bodies (ABs) in *G. theta*

During experiments with *G. theta* cultures grown in standard conditions, we observed the presence of extracellular vesicles in solution (the extracellular space). In stationary and death phases of the culture, when the cells experienced nutrient depletion and aging, the proportion of these vesicles (hereafter *G. theta* Apoptotic Bodies (Gt-ABs)) increased and was accompanied by the progressive decrease in cell number. As illustrated in Fig. 1a, the concentration of Gt-ABs in the culture's population increased over time, with the highest number during the death phase. Although Gt-ABs were also present in the exponential growth phase, their concentration was very low and steady ($0.2 \times 10^6$ counts/mL), significantly different (Brown-Fortsythe and Welch's ANOVA followed by Dunnett's T multiple comparisons test) compared to the stationary ($0.48 \times 10^6$ counts/mL) and death phases ($0.75 \times 10^6$ counts/mL) of the culture.

Gt-ABs were smaller (2−5 μm) than the *G. theta* cells (5−14 μm), and fluorescence microscopy as well as transmission electron microscopy (TEM) confirmed their differences in size and ultrastructural complexity (Supplementary Fig. 1). These analyses showed that Gt-ABs lacked the distinctive structural features of *G. theta* cells such as flagella, intact organelles (e.g., chloroplasts) and photosynthetic or accessory pigments. Consequently, they did not display any chlorophyll autofluorescence signal (photosynthetic efficiency (Fv/Fm); Supplementary Fig. 1b), which, in contrast, was measurable and constant throughout the experiment in living cells (0.65 in exponential growth phase of the population, 0.67 in stationary phase, and 0.62 in death phase). Cell death increased with culture age, as evaluated by SYTOX green staining (Fig. 1b). During exponential growth of the population the percentage of dead cells was 13.9%, rising to 17.2% during stationary phase and reaching 36.8% by the end of the death phase (day 42). Using terminal deoxynucleotidyl transferase dUTP nick end labeling (TUNEL) staining, we determined that on average 46.9% of the cells in the culture were experiencing apoptosis in the early death

phase (day 28; Fig. 1b). Applying Fluorescence Lifetime Imagine Microscopy (FLIM) we successfully differentiated the TUNEL (fluorescein) fluorescence signal from the autofluorescence in *G. theta* cells, with intensity-weighted mean fluorescence lifetime value of 3.9 ns and 0.6 ns, respectively. At the same time, the expression of the two metacaspase genes *GtMCA-I* (fold change (FC) = 4.12) and *GtMCA-III* (FC = 4.71) significantly increased (Brown-Fortsythe and Welch's ANOVA followed by Dunnett's T multiple comparisons test) during death phase, with higher transcription levels at day 28 (Fig. 1c). The expression of metacaspases, the structural homologs to metazoan caspases, has been linked to PCD. Transcription of both metacaspase genes was low during exponential and stationary growth of the *G. theta* population (FC ≈1.22).

### *G. theta* cells display the ultrastructural markers of apoptosis and AB production

To characterize our finding that *G. theta* cells produce ABs during apoptosis, we investigated aging *G. theta* cultures (unfiltered and filtered to enrich Gt-ABs) by TEM (Fig. 2). We compared the observed features to the ultrastructural markers common in metazoan apoptotic cells. In stationary and death phases of the population, *G. theta* cells displayed an apoptotic phenotype similar to metazoan apoptotic cells, consisting of chromatin condensation and/or nuclear fragmentation (Fig. 2a, c), production of Gt-ABs inside the cells (Fig. 2a, b), formation of small surface membrane blebbing and large dynamic blebs (Fig. 2b, d) and detachment of Gt-ABs from the cells (Fig. 2c) leaving an empty space in the cytoplasm (Fig. 2d). Extracellular, free-floating Gt-ABs from filtered, enriched cultures displayed an electron dense composition, encapsulated by the cell's periplast layer alongside fragments of cytoplasm and shrunken organelles (Fig. 2e). Importantly, cell integrity was maintained due to the conservation of the plasma membrane and the inner periplast layer. Mitochondria, chloroplasts and Golgi apparatus were maintained inside apoptotic *G. theta* cells but there were different degrees of shrinkage and/or fragmentation. Gt-ABs remained in solution for a long time. In the final stages of apoptosis, all these events led to cell dismantling and eventually, culture demise.

### Pharmacological induction of apoptosis

Next, we investigated the effect of apoptotic inducers staurosporine (STS) and carbonyl cyanide m-chlorophenylhydrazone (CCCP) on *G. theta* cells (Supplementary Fig. 2). Treatment with the pan-kinase inhibitor STS led to cell decline and generation of Gt-ABs in the culture within 4 h. Similar results were observed when the mitochondrial uncoupler CCCP was added in the culture. The number of Gt-ABs showed a 4-fold increase during 6 h of treatment, while there was no cell death and Gt-ABs formation in the control culture after adding only the vehicle (Supplementary Fig. 2a). Time-lapse differential interference contrast (DIC) microscopy revealed the morphological steps of the disassembly process in STS and CCCP treated *G. theta* cells (Supplementary Fig. 2b). Cell rounding, surface blebbing and dynamic blebbing were observed at about 2.5 h and 3.5 h after the addition of 5 μM STS and 49 μM CCCP, respectively. The apoptotic phenotype observed was identical with the one of an aged cell, undergoing spontaneous apoptosis. Collectively, these data suggest disassembly of *G. theta* cells through an apoptotic mechanism.

### Classification and characterization of Gt-ABs by FACS

The morphologic and phenotypic differences between Gt-ABs and cells (Supplementary Fig. 1) aided in the identification of distinct Gt-AB populations in stationary and death phases according to their relative size (subpopulations of approximately 2 μm and 3−5 μm) and their 4´,6-diamidino-2-phenylindole (DAPI) fluorescence signal (DAPI-positive: containing DNA, and DAPI-negative: not containing DNA). Consequently, using a sequential gating strategy (Fig. 3a(i)), within the DAPI (+/-) subpopulation (Fig. 3a(iv)) we selected four Gt-AB subpopulations per

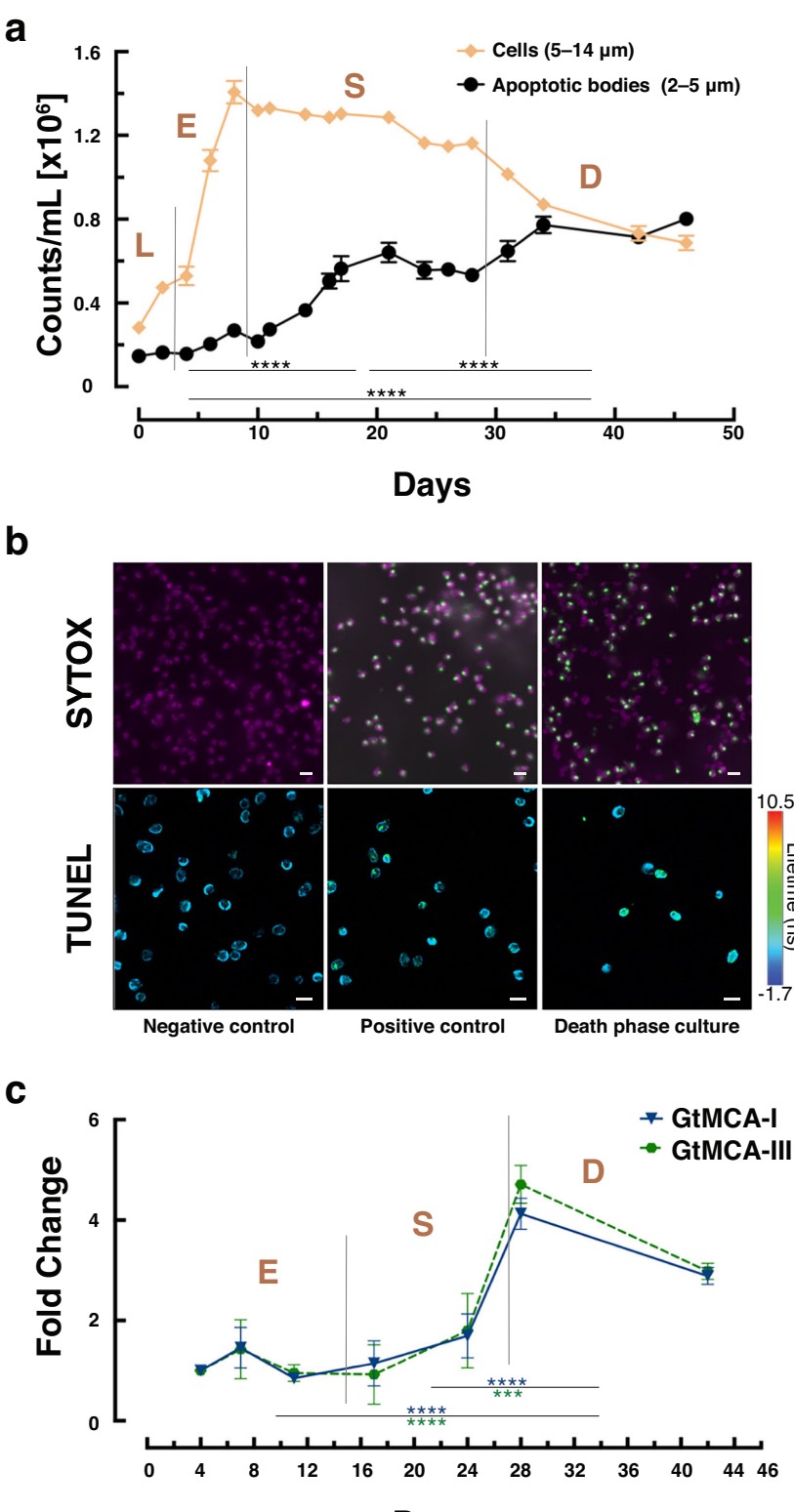

culture age: small DAPI-positive Gt-ABs (relative size ca. 2 μm), small DAPI-negative Gt-ABs, large DAPI-positive Gt-ABs (relative size ca. 3–5 μm) and large DAPI-negative Gt-ABs (Fig. 3a(ii), (v), (vi)). The Gt-ABs were sorted in a single-particle format. The population of *G. theta* cells was identified by their relative size (ca. > 5 μm) and their chlorophyll autofluorescence emission, and was therefore excluded from sorting

(Fig. 3a(ii), (iii)). Using confocal microscopy, we verified the absence of *G. theta* cells, the presence of the identified Gt-ABs subpopulations and validated their estimated sizes (Fig. 3b).

FACS analyses further showed that the percentage of total Gt-ABs sorted changed in relation to culture age. The relative proportion of small Gt-ABs within the ABs population (non-cell fraction)

**Fig. 1 | Growth and regulated cell death of the *G. theta* population over time.**
**a** Growth of the *G. theta* cell population and production of ABs in standard conditions. Dotted lines indicate the transitions between the growth phases, from left to right: lag (L), exponential (E), stationary (S), and death (D) culture phases. Samples for subsequent analysis were taken in the exponential phase (days 4 and 7), at the onset of the stationary phase (day 11), during early and late stationary phase (days 17 and 24), and during early and late death phase (days 28 and 42). An evident decline in cell number was observed from stationary to death phase, while the concentration of ABs increased with culture age. The means of the counts of the Gt-ABs for each growth phase were compared using Brown-Forsythe and Welch's ANOVA, followed by Dunnett's T multiple comparisons test (****$P < 0.0001$). Non-significant comparisons with $P > 0.05$ are not shown. Data are presented as mean values +/− standard errors of mean (SEM) of biologically independent samples ($n = 4$) and two technical replicates per sample. **b** Cell viability assays using SYTOX green (upper panels) and TUNEL (lower panels) demonstrating the increased

numbers of dead cells during death phase. SYTOX (green) labels dead cells, while living cells retain their chlorophyll autofluorescence (magenta); scale bars correspond to 10 µm. Representative micrographs from 3 biological replicates and 3 independent experiments are shown. TUNEL (fluorescein, green) labels apoptotic cells relative to living cells (chlorophyll autofluorescence, blue), scale bars correspond to 20 µm and the lifetime color bar corresponds to the range in nanoseconds (−1.7 to 10.5 ns). Representative micrographs from 2 independent experiments are shown. **c** Metacaspase proteases associated with cell death in non-metazoan organisms, showing a significant increase in gene expression (FC) during the early death phase. The means of the FC for each growth phase were compared using Brown-Forsythe and Welch's ANOVA, followed by Dunnett's T multiple comparisons test (***$P = 0.004$, ****$P < 0.0001$). Data are presented as mean values +/− SEM of biologically independent samples ($n = 3$) and two technical replicates per sample. Non-significant comparisons with $P > 0.05$ are not shown. Source data are provided as a Source data file.

---

more than doubled between stationary phase (27%, day 17) and death phase (69%, day 42). In contrast, the proportion of large Gt-ABs significantly decreased from 73% (day 17) to 31% (day 42) (two-way ANOVA; significant interaction between relative sizes and growth phases; Fig. 3c). DAPI-negative subpopulations were significantly more predominant than DAPI-positive subpopulations in both stationary and death phases, as confirmed by confocal microscopy (Fig. 3b) and FACS (Supplementary Table 1, number of recorded events within the ABs population). The significant differences were due to the significant interaction (two-way ANOVA) between both factors simultaneously, the DAPI signal (+/−) in the Gt-ABs and the growth phases and within each subpopulation of Gt-ABs (two-way ANOVA).

## Biochemical overview of Gt-ABs using confocal Raman spectroscopy (RS)

Sorted Gt-ABs were analyzed by RS (Fig. 4a). Imaging mode was used so that every spectrum of the recorded hyperspectral maps described the biochemical features of individual Gt-ABs in random clusters. Figure 4b shows a comparison of representative mean Raman spectra of Gt-ABs harvested from the *G. theta* culture in stationary and death phase, respectively, separated and sorted into the four subpopulations by FACS.

All spectra displayed peaks in ranges frequently reported to correspond to major constituents of biological samples. Bands assigned to nucleic acids (668–823 cm$^{-1}$) and proteins (1531–1641 cm$^{-1}$ and 1684–1697 cm$^{-1}$) had the largest contribution, followed by carbohydrates (370–430 and 479–498 cm$^{-1}$), while lipids (1750–1771 cm$^{-1}$) had the smallest contribution to the spectral profile of Gt-ABs. Protein peaks displayed the highest intensities in all samples, independent of the culture phase. They originated from the amide I band of the peptide bond and signals previously associated to individual amino acids (phenylalanine (Phe), tyrosine (Tyr), tryptophan (Trp), proline (Pro), valine (Val), methionine (Met) and glutamate (Glu, ionic))[26–32]. Peaks originating from nucleic acids were found in both DAPI-positive and -negative samples; these peaks displayed distinctive ring breathing modes and phosphate backbone vibrations typical for nucleic acids[26, 28–31,33–35]. More prominent band shapes and relative peak intensities were observed in DAPI-positive Gt-ABs collected during the death phase of the culture's population. In these DAPI-positive samples distinctive peaks resulting from DNA and RNA were detected, while DAPI-negative Gt-ABs only featured bands assigned to DNA.

On average, the spectra of DAPI-negative samples displayed more peaks across the entire spectral range, with higher general intensities than spectra of DAPI-positive samples. Spectra originating from three Gt-AB subpopulations differed between the stationary and the death phases. The small DAPI-positive and large DAPI-positive had higher

intensities in the death phase. The large DAPI-negative subpopulations showed the opposite; the higher intensities were observed in the stationary phase. In contrast, the spectra of small, DAPI-negative Gt-ABs did not differ between these culture phases.

## A closer look into Gt-ABs: What happens during stationary and death phase of the *G theta* population?

Principal Component Analysis (PCA) was used to reduce the dimensionality of the Raman spectroscopic data ($n = 99$). The model represented the entire dataset by 10 Principal Components (PC), of which the first two collectively described more than 50% of the total variation (PC1 37.5% plus PC2 14.6%). The PCA Scores plot mainly revealed the grouping between Gt-ABs in terms of size and DAPI fluorescence (Supplementary Fig. 3). Based on these results, Orthogonal Projections to Latent Structures–Discriminant Analysis (OPLS–DA) was performed to identify the source of variation between and within the groups.

This supervised classification model highlighted the differences between the Gt-AB subpopulations in relation to the two culture phases. The distinct separation between subpopulations originating in the stationary or death phases of the culture is illustrated on small DAPI-positive Gt-ABs (Supplementary Fig. 4). OPLS–DA models with the highest cumulative Q2 values (indicating the most significant differences between growth phases) were considered and compared: small DAPI-positive (Q2 = 0.538; Fig. 5a); large DAPI-positive (Q2 = 0.899; Fig. 5b) and large DAPI-negative (Q2 = 0.493; Fig. 5c). From the selected models, only the Loadings corresponding to peaks with ≥ 50% correlations were assigned for the biochemical characterization of Gt-ABs during apoptosis (Fig. 5d–f).

Most of those peaks ($n = 71$) could be assigned to a specific biochemical component, but some ($n = 27$) had ambiguous assignments due to the non-specificity of the underlying molecular moieties of the Raman signatures (Table 1 summarizes peak assignments; a list of features, associated peak assignments from literature and statistical correlations are given in Supplementary Data 1). The assigned peaks were grouped into different categories: DNA, RNA, proteins, lipids and carbohydrates. Due to ambiguity, joint categories, such as DNA/RNA, DNA plus proteins, DNA plus lipids, DNA/RNA plus proteins, DNA/RNA plus lipids, proteins plus lipids and lipids plus carbohydrates were also used (Table 1, Fig. 6). Overall, most of the spectral peaks, in terms of their total integrated areas, were assigned to DNA and proteins and hence these biochemical molecules are considered the dominant components of Gt-ABs. However, while the proportion of DNA and proteins was lower in Gt-ABs of the stationary culture phase (11.7% and 23.6%, respectively) than of the death culture phase (26.2% and 40.2%, respectively), the percentage of lipids was similar in both culture phases (12.3% in stationary; 11.6% in death phase). Carbohydrates were present in Gt-ABs from the stationary phase (18.3%) but were not detected in Gt-ABs from the death phase.

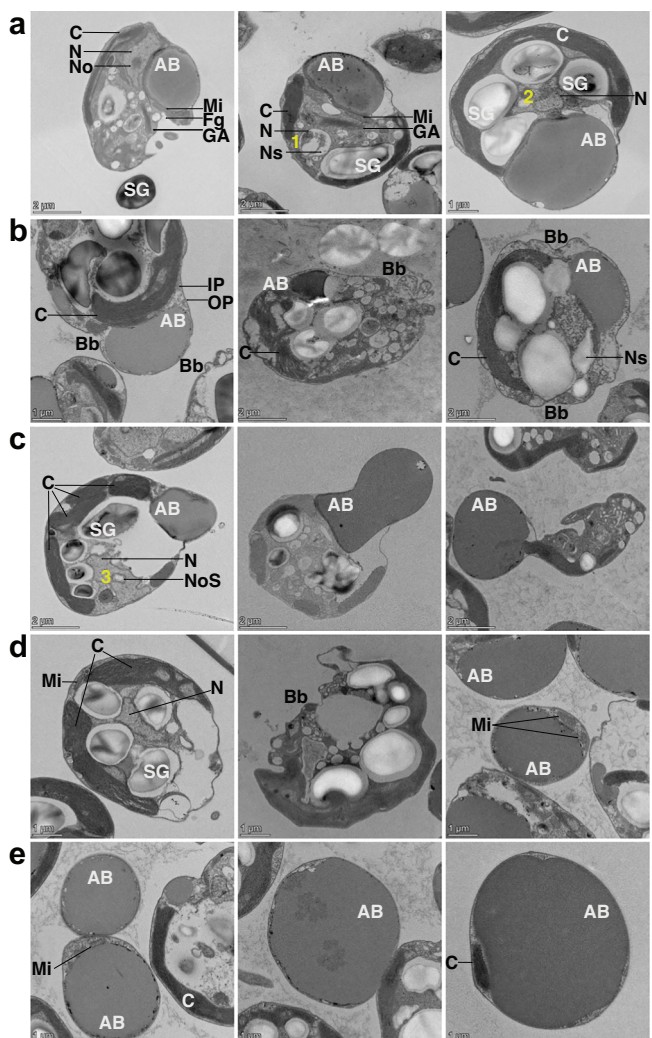

**Fig. 2 | TEM analyses to display ultrastructural changes of *G. theta* cells during apoptosis and AB production.** Rows: **a** Production of Gt-ABs inside apoptotic cells. **b** Formation of small surface membrane blebbing (Bb) and large dynamic membrane blebbing characteristic of apoptosis progression. **c** Detachment of Gt-ABs from the cells. **d** Cytoplasmic space left by the Gt-ABs after detachment along with membrane blebbing. **e** Extracellular Gt-ABs from filtered, enriched cultures, free floating in solution, displaying their detailed ultrastructure with an electron dense composition and covered by the periplasmic layer of the cell, also encapsulating fragments of cytoplasm and organelles. Hallmarks of apoptosis and cell disassembly are identified as follows: nuclear shrinkage, chromatin condensation and fragmentation (1); nuclear shrinkage, chromatin condensation and marginalization (2); nucleolus degradation (3); Bb − membrane blebbing; AB − apoptotic body. Different organelles and cellular components can be appreciated in all images: N − nucleus, Ns − nuclear space, No − nucleolus, NoS − nucleolus space, C − chloroplast, GA − Golgi apparatus, Mi − mitochondrion, Fg − flagella, IP − inner periplast, OP − outer periplast, SG − starch granule. Representative micrographs from 6 independent experiments are shown.

## Discussion

### Apoptosis and apoptotic body production in *G. theta*

Apoptosis is believed to have evolved about 1 billion years ago, together with the evolution of multicellular organisms. However, here we describe for the first time apoptosis in a non-metazoan organism, a photosynthetic, unicellular alga that arose from secondary endosymbiosis before multicellularity. While metazoan apoptosis is well studied − animal cells undergo ordered remodeling during apoptosis, where the production of metazoan ABs is a crucial and definitive hallmark of the process[6–8,36,37] − AB production

has not been reported previously for other multi- or unicellular organisms.

Nutrient limitation and constant environmental changes lead to stagnated growth, senescence and death of natural microalgal populations in aquatic ecosystems[11]. To maintain a healthy culture with biomass production, PCD is believed to be a key process, necessary for the biogeochemical cycles in oceans[10,25]. In our experiments, *G. theta* cultures aged in standard growth conditions and experienced nutrient deprivation, evidenced by the stagnation of biomass production. However, we did not observe a collapse of the culture, not even during its death phase: the photosynthetic efficiency of the cells remained in the normal range described for cryptophytes[24]. Additionally, SYTOX green staining, which labels unfragmented DNA in dead cells with increased membrane permeability[2], was low (36.8%). Apoptotic cells are known to continue to function and can retain their membrane[3,36], which could explain the low percentages of SYTOX green straining observed in our experiments. Nonetheless, we detected apoptotic DNA fragmentation in nearly 50% of the cells in the culture during the death phase, confirmed by TUNEL[38,39] experiments. Additionally, we observed the presence of extracellular, free-floating vesicles in the medium, with sizes ranging from 2 to 5 μm. They resembled metazoan ABs[40], which are membrane-bound extracellular vesicles of similar size (1−5 μm), produced by certain types of animals cells[7,37,41] and cell lines[42] during apoptosis. We found that the concentrations of these vesicles (Gt-ABs) significantly increased from the exponential culture phase to the death phase (Fig. 1a), connecting their release with apoptosis. Instead of a culture collapse, we therefore witnessed a few *G. theta* cells dying in an ordered manner, potentially aiding culture maintenance and fitness of the remaining cells[10].

### Ultrastructural and subcellular markers of apoptosis and AB production

We used TEM to investigate *G. theta* cells for PCD markers recognized in metazoan apoptosis (Fig. 2), such as cell rounding and volume decrease, chromatin condensation, nuclear fragmentation, the formation of membrane blebbing, the production of ABs and the fragmentation and demise of the cell[4,7,43]. Whereas *G. theta* cells growing in the exponential phase displayed well defined nuclear membranes as well as chromatin localized in the nucleolus (Supplementary Fig. 1c, e), during apoptosis we observed chromatin condensation and its migration to the nuclear envelope (Fig. 2a, c). DNA fragmentation and complete degradation became apparent with the loss of nuclei and/or the nucleolus. In metazoans, similar features are very well described and mark the early stages of apoptosis[41,44]. They are also well documented in the chlorophyte alga *Dunaliella tertiolecta* during PCD induced by light deprivation[14]. The morphological changes observed in the nucleus of *G. theta* cells during early apoptosis correlated well with the DNA degradation detected by TUNEL in the culture death phase.

The absolute hallmarks of metazoan apoptosis are the formation of plasma membrane blebbing and the production of ABs. Together with the fragmentation and demise of the cell, these constitute a process called apoptotic cell disassembly[7]. Indeed, in *G. theta* we observed membrane blebbing and the production of apoptotic bodies. Their sizes (2−5 μm) were like metazoan ABs (1−5 μm)[7,41,45], but distinctively smaller than *G. theta* cells (5−14 μm). Using time-lapse imaging we documented cell rounding, apoptotic membrane blebbing and Gt-ABs formation in both pharmacologically induced apoptosis and naturally aged *G. theta* cells (Supplementary Fig. 2b). Furthermore, these Gt-ABs lacked chlorophyll and other pigments (Fig. 1b, Supplementary Fig. 1a, b, g, h), while − independent of the culture state − *G. theta* cells contain photosynthetic pigments, evidenced by chlorophyll autofluorescence, a measure of the efficiency and regulation of photosynthesis[46]. Although we observed Gt-ABs encapsulated with remnants of once functional organelles, such as the chloroplast, the lack of chlorophyll autofluorescence in Gt-ABs indicated their lost

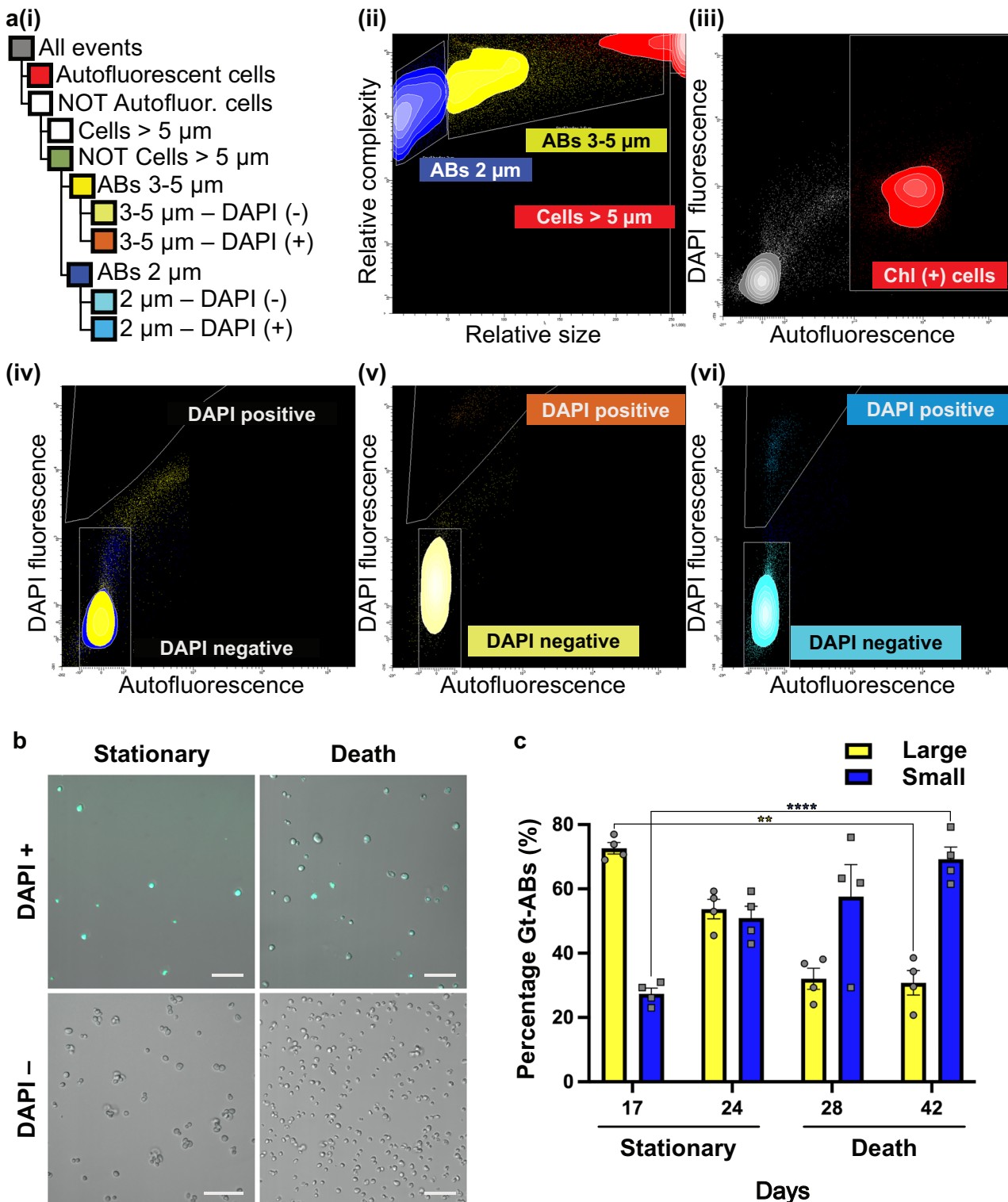

functionality. We also observed mitochondria inside Gt-ABs (Fig. 2e). Metazoan ABs have also been reported to contain residual organelles, such as mitochondria and endoplasmic reticulum[7,41,45]. While studies about the functionality of these metazoan organelles are lacking, in *G. theta* cells shrunken and/or fragmented organelles are most likely randomly packed inside ABs during cell dismantling.

Apoptosis in metazoans is initiated and executed by caspases, cysteine proteases of the C14 family, which cleave their substrate after an aspartic acid residue[47]. Despite the indisputable importance of these proteases in metazoa, caspases have not been detected in other

eukaryotes. Instead, structurally highly homologous proteins to caspases have been identified and termed metacaspases[48]. Although metacaspases share the structural p20 and p10 domains of caspases, their substrate specificity differs from caspases. Based on structural differences, metacaspases are sub-divided into types I, II, and III. *G. theta* contains one type I metacaspase (Gt-MCA-I) and one type III metacaspase (GtMCA-III)[49]. In physiological conditions during aging and death of the *G. theta* population, the gene expression of both metacaspases rose within the stationary culture phase and peaked in the early death phase, supporting their involvement in apoptosis.

**Fig. 3 | Identification, sorting and isolation of Gt-AB subpopulations using FACS. a** Gating strategy (**i**). The identified population of algal cells was excluded from sorting (using its chlorophyll autofluorescence and relative size > 5 μm) (**ii**, **iii**). Gt-ABs were then identified in the subsequent population based on their estimated relative size of 2–5 μm (**ii**). Within these, the subpopulation DAPI-positive Gt-ABs was selected and compared with the control unstained sample (**iv–vi**). This resulted in four subpopulations of Gt-ABs distinguished, analyzed, and sorted based on relative size (2 μm and 3–5 μm) and DAPI fluorescence signal (**ii**, **v**, **vi**). **b** As shown in the images for the large (3–5 μm) Gt-AB subpopulation, all the sorted Gt-AB subpopulations were confirmed for size, DAPI fluorescence signal and absence of chlorophyll autofluorescence, using confocal microscopy (one fluorescence track for the simultaneous detection of DAPI and chlorophyll fluorescence). DAPI-negative subpopulations were more predominant than DAPI-positive subpopulations (Supplementary Table 1). The DAPI-negative subpopulation of the

stationary phase (left, lower panel; scale bar 10 μm) was visualized with a higher resolution objective (see Methods) for an enhanced overview, other scale bars correspond to 20 μm. Representative micrographs from 2 independent experiments with similar results are shown. **c** The percentage of total Gt-ABs sorted within the ABs population in stationary and death growth phase was recorded per sampling day. Data are presented as mean values +/− SEM of biologically independent samples ($n = 4$). The differences in the proportions of small Gt-ABs and large Gt-ABs were significant due to the interaction between the relative sizes and the growth phases ($P < 0.0001$). The analysis was performed with two-way ANOVA. Follow-up multiple unpaired t tests confirmed the differences in the means between the two growth phases within each group, large Gt-ABs (****$P = 0.000007$) and small Gt-ABs (**$P = 0.004792$). Non-significant comparisons with $P > 0.05$ are not shown. Source data are provided as a Source data file.

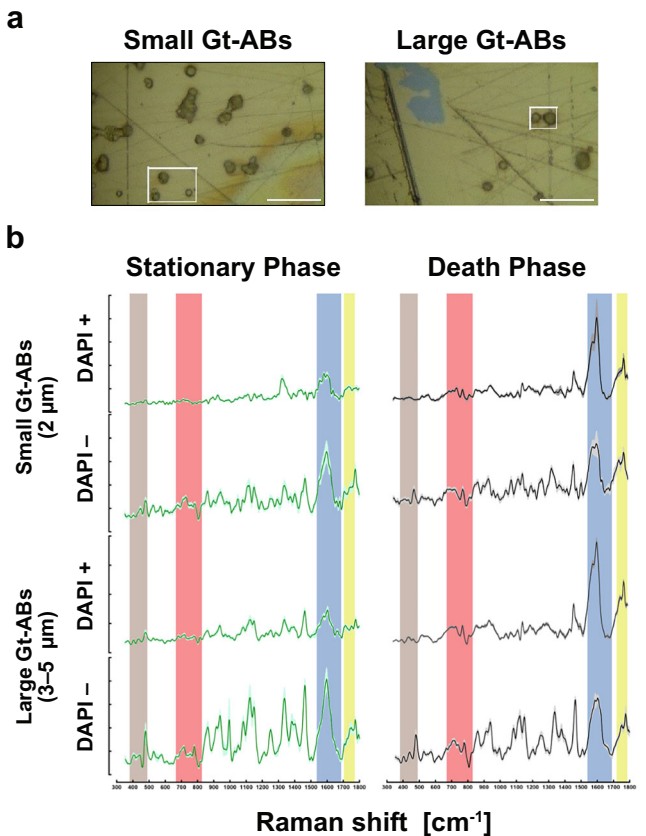

**Fig. 4 | Raman fingerprint of Gt-ABs. a** Confocal RS images representative of the 2 μm and 3–5 μm Gt-AB subpopulations. Every spectrum describes the biochemical features of individual Gt-ABs deposited and grouped on the gold mirror. Randomly selected areas were imaged. Scale bar corresponds to 20 μm. **b** Average Raman spectra obtained using an excitation wavelength of 405 nm and 10 s of exposure for 2 accumulations for each spectrum. The solid lines correspond to an average of at least 8 spectra per ABs subpopulation per growth stage ± the SEM (shaded areas). Green spectral lines correspond to ABs harvested in the stationary phase and black lines correspond to ABs of the death phase. Ranges of the spectral peaks are frequently reported and correspond to major constituents of biological samples. The Raman peaks indicating carbohydrates are highlighted in brown (370–430 cm⁻¹), nucleic acids in red (668–823 cm⁻¹), proteins in blue (1531–1641 and 1684–1697 cm⁻¹) and lipids in yellow (1750–1771 cm⁻¹). Peak intensity is given at the Y-axis (a.u.). Source data are provided as a Source data file.

Metacaspases of microalgae have been linked to PCD[15,18]. In addition, the pharmacological compounds STS and CCCP, known to trigger apoptosis and ABs formation in metazoan cells[50,51], induced an apoptotic cell morphology in *G. theta*. This suggests that a pathway analogous to the metazoan mechanisms underlying apoptosis exists in *G.*

*theta*. Like animal cells, *G. theta* cells possess fluid cell envelopes with structural plasticity, that appear to be involved in the execution of cell disassembly during apoptosis. In *G. theta* the plasma membrane is protected by inner and outer periplast layers. The inner layer is made of epiplastin, a proteinaceous substance that provides additional flexibility and protection to the cell membrane[24]. Therefore, like metazoa, *G. theta* has the structural flexibility to form membrane blebs and membrane-derived vesicles (e.g., ABs; Fig. 2, Supplementary Fig. 2b). By contrast, chlorophyte algae and plants (green linage) are surrounded by rigid cell walls[8], diatoms by silica walls and dinoflagellates are covered by cellulose plates[20]. In conclusion, *G. theta*'s apoptotic morpho-phenotype is highly identical to metazoan apoptosis. Moreover, TEM images of ABs[37,42] derived from thymocytes, human mast cells, murine cells and erythroleukemia cell lines show nearly identical ultrastructures to Gt-ABs.

Despite all these similarities between metazoan ABs and Gt-ABs described above, apoptotic cell disassembly in *G. theta* seems to be simplified compared to metazoan apoptosis, as expected for an unicellular organism: (i) Gt-ABs remained inside the cells, while small and dynamic blebs formed (Fig. 2a, b); (ii) When Gt-ABs were released into the extracellular space, the cells maintained their integrity (Fig. 2c, d) and Gt-ABs remained in solution for long time periods (Fig. 2e), while in metazoan the apoptotic cells fragment and ABs are quickly removed by phagocytosis; (iii) The generation of apoptotic membrane protrusions (apoptopodia or beads on a string), an apoptotic feature in certain metazoan cells[1] and cancer cell lines[7], was lacking in *G. theta*. Apoptosis in *G. theta* progressed concomitantly with nutrient deprivation, finally leading to cell fragmentation and culture demise.

Other types of extracellular vesicles (EVs) have been reported in metazoan and non-metazoan organisms, plants and bacteria (Supplementary Data 2). In comparison, ABs are larger (size range ca. 1–5 μm) than the heterogeneous populations of other types of EVs (ca. 20–150 nm and microvesicles up to 1 μm). Some EVs are membrane bound, as shown in the cyanobacteria *Prochlorococcus sp.*[52], while ABs are released from the cells and remain "free-floating" in the extracellular space. So far, ABs are the only type of EVs that package shrunk/fragmented organelles and are a product of cell disassembly during cell death.

## Biochemical characterization of Gt-ABs, insights into ageing and death

Confocal RS is frequently applied to study biological systems in vivo/in situ due to its low sensitivity to water, thus avoiding water interference with the active biomolecular signals. Metazoan ABs have been characterized by other fluorescence microscopy techniques and protein markers[7,38], but not by RS. Instead, RS has been applied to animal cell lines undergoing apoptosis[34] or other types of EVs[26]. In algae, RS is mainly used to identify and quantify industry-relevant biomolecules[27,53,54]. Here, for the first time, we provide a biochemical "snapshot" of ABs using RS.

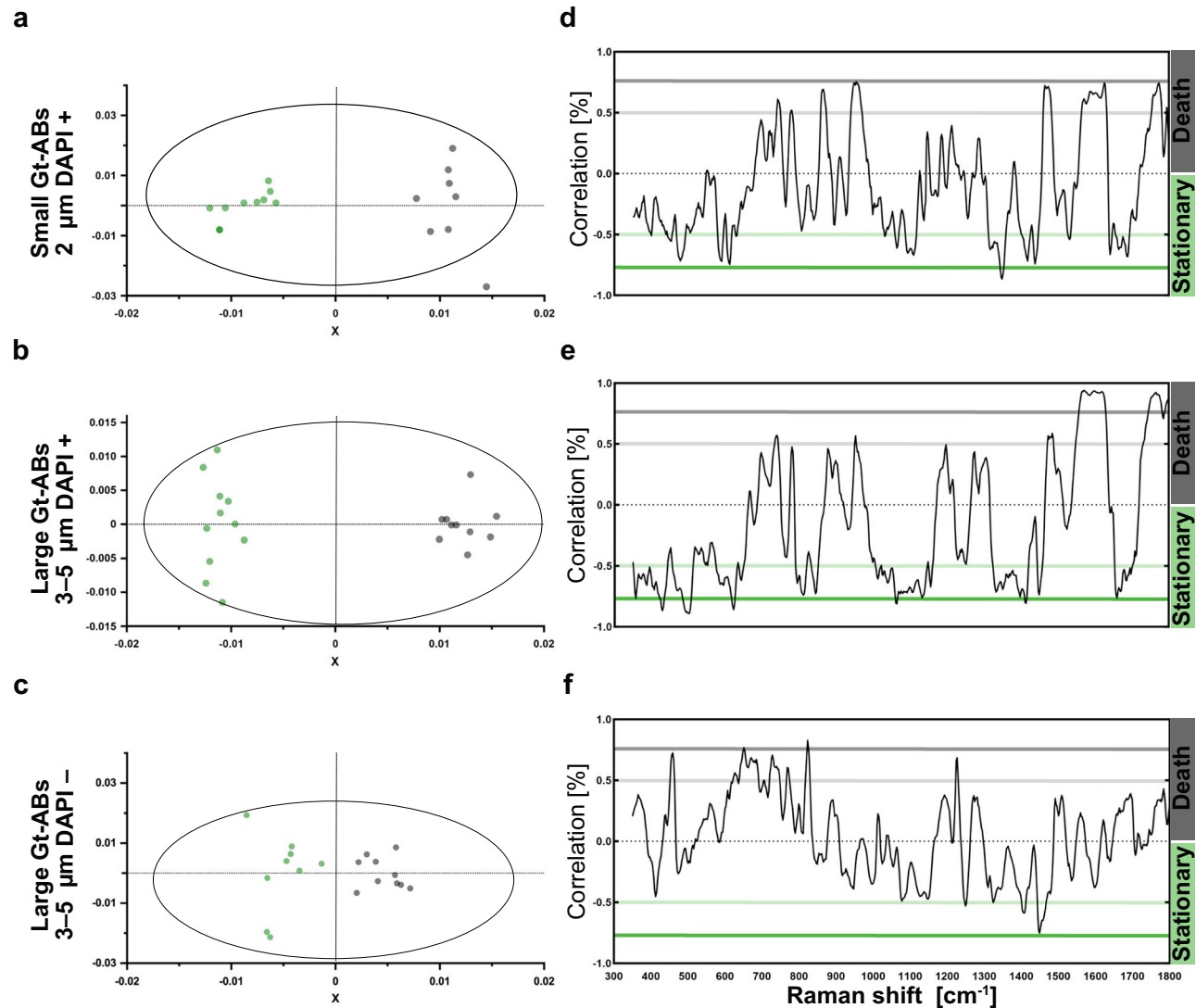

**Fig. 5 | Comparisons between selected Gt-ABs of the stationary and death culture phases, based on orthogonal projections to latent structures−discriminant analysis (OPLS−DA) subpopulations with the highest Q2 values.** Top row: 2 μm, DAPI-positive (Q2 = 0.538, N = 18), middle row: 3−5 μm, DAPI-positive (Q2 = 0.899, N = 21) and bottom row: 3−5 μm, DAPI-negative (Q2 = 0.493, N = 19). **a**−**c** Score plots, where the X-axis is the first predictive component and the Y-axis is the first orthogonal component. Gt-ABs in the stationary culture phase are displayed by green circles and Gt-ABs in the death culture phase by black circles. The ellipsoid indicates the Hotelling 95% confidence interval. **d**−**f** corresponding correlation-scaled Loading plots. Peaks with 50% to 75% correlation (located between the soft and solid colored lines) and peaks with more than 75% correlation (beyond the solid colored lines) were considered significant and were included in peak assignments to determine the chemical content of the Gt-ABs (Table 1 and Supplementary Data 1). The colors of the lines indicate whether the peak is more intense in the stationary (green) or in the death (black) phase of the culture. Source data are provided as a Source data file.

Fluorescence peaks from chlorophyll, carotenoids, or other light-harvesting pigments frequently reported in the Raman spectra of algal cells[27,28,35,53], could potentially hamper RS analyses. However, these molecules were absent in Gt-ABs as determined by fluorescence microscopy and FACS. None of the collected Raman spectra showed the pronounced background commonly caused by pigment fluorescence in algal cells. We employed a 405 nm laser to minimize potential carotenoid interference. Carotenoids are resonantly enhanced in RS, especially with 488−532 nm laser excitations, and even in the near infrared range (785 nm)[54]. None of the spectra contained the intense signature peaks of β-carotene at 1008 cm$^{-1}$ (ρ(CH$_3$)), 1150 cm$^{-1}$ (vs (C−C)) and 1520 cm$^{-1}$ (vs(C = C))[28,53].

ABs have never been reported in phytoplankton before, but metazoan ABs are known to pack genetic material, such as DNA and RNA, either fragmented or intact[1,7,41]. We identified DNA as well as RNA in DAPI-positive samples, while DAPI-negative Gt-ABs only contained DNA. Surprisingly, DAPI-negative Gt-AB subpopulations were

significantly more abundant than the DAPI-positive ones (Supplementary Table 1). However, since *G. theta* genes are GC rich[55], single stranded or fragmented DNA in the Gt-ABs are less likely to bind DAPI, which preferentially binds to A-T rich chromosomal regions and double stranded DNA[56]. Thus, to avoid bias in the biochemical characterization of the Gt-ABs with RS, DAPI-negative subpopulations were also sorted and included in this study. ABs from animal cell lines[57] have been found to contain either DNA or RNA, alternatively both nucleic acid molecules can be co-distributed into the same ABs[7]. Due to the high sensitivity of Raman spectroscopy for nucleic acids[58], these were detected in all subpopulations of Gt-ABs, including the DAPI-negative (Table 1, Supplementary Data 1). In contrast to metazoan ABs[4,7,57], Gt-ABs lacked substantial amounts of RNA (few, low intensity peaks). OPLS−DA analysis highlighted peaks with high correlation, assigned to distinctive features of RNA (RNA phosphodiester bands at 811 and ribose vibration at 861 cm$^{-1}$; Table 1, Supplementary Data 1). We found the ribose vibration triad reported in biological tissues (867, 915 and

**Table 1 | Peak assignments describing the contents of Gt-AB subpopulations using significant OPLS-DA models**

| ASSIGNMENTS | | PEAKS [cm⁻¹] in Gt-ABs subpopulations | | |
|---|---|---|---|---|
| | | Size 2 µm-DAPI positive | Size 3–5 µm-DAPI positive | Size 3–5 µm-DAPI negative |
| SINGLE | DNA | | | |
| | Ring breathing modes NA bases, Thymine, Guanine, Adenine, Cytosine, O-P-O stretch, Deoxyribose vibration | **744**, 1094, 1418, **1462**, 1671 | 794, 1091, 1416, **1485**, 1671 | **668, 677, 679, 725, 737, 742, 823** |
| | RNA | | | |
| | Ribose vibration, C5'-O-P-O-C3' phosphodiester bonds | **861** | 811 | |
| | PROTEINS | | | |
| | Aminoacids (Phe, Met, Tyr, Pro, Val, Glu), α-helix, COO⁻ vib, C = C stretch, peptide backbone, Amide I (β- sheet), Amide I (disordered-non hydrogen bonded), Amide I (turns and bands) | 452, 612, 629, **947, 952**, 1031, 1531, **1566, 1601, 1623**, 1641, 1686 | **629**, 644, **950**, 1010, 1027, 1038, 1156, 1398, **1566, 1599, 1619**, 1684, 1691, 1697 | **456, 624, 651**, 1407 |
| | LIPIDS | | | |
| | Cholesterol, Phosphatidylinositol, Phosphatidylserine, Fatty acids (FA) | 588, 1064, 1438, **1756, 1762, 1771** | 429, 586, 593, 1064, **1762** | |
| | CARBOHYDRATES | | | |
| | Starch, Glycogen, Trehalose, Glucosides (degraded glusose), polyssacharides | 400, 479, 486 | 373, 387, 402, 407, 471, 498, 844, 1105 | |
| AMBIGUOS | DNA/RNA | | | 768 |
| | DNA plus PROTEINS | **1471, 1584**, 1665 | 622, **1474, 1480** | |
| | DNA plus LIPIDS | | **735**, 1380 | |
| | DNA/RNA plus PROTEINS | 1317, 1319, 1348 | 1333, 1351, 1365, 1371 | **1224**, 1249 |
| | DNA/RNA plus LIPIDS | **775** | 1073, 1079, 1132 | |
| | PROTEINS plus LIPIDS | 612 | 1449, 1660 | 1449 |
| | LIPIDS plus CARBOHYDRATES | 1110 | | |

Raman bands in bold represent peaks in the death phase of growth.

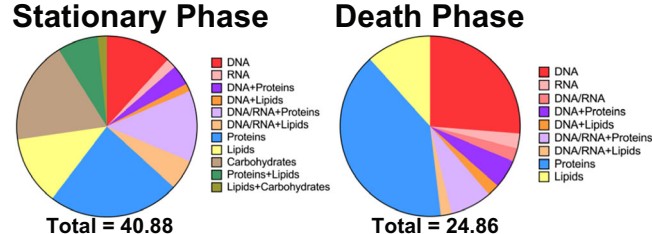

**Fig. 6 | Gt-AB contents, distributed according to Raman peak intensities, in relation to culture phase.** The biochemical characterization of Gt-ABs during apoptosis was performed using peak intensities in the Raman spectra. Only peaks with correlations ≥ 50% were used for the analysis. Raman signatures that allowed assignments to a single biochemical molecule were classified as: DNA, RNA, proteins, lipids or carbohydrates. Peaks with ambiguous assignments were categorized as: DNA/RNA, DNA plus proteins, DNA plus lipids, DNA/RNA plus proteins, DNA/RNA plus lipids, proteins plus lipids, lipids plus carbohydrates. Bands assigned to DNA and proteins had the largest contribution to the spectra and hence they dominate the content of Gt-ABs. Proportions of the Gt-AB contents changed during apoptosis from stationary to death phase of the population. Source data are provided as a Source data file.

974 cm⁻¹, correlation <50%)[30,31] in Gt-ABs at 867, 917 and 973 cm⁻¹, respectively. RNA signals could be detected in DAPI-positive Gt-ABs in both stationary and death phases, but the spectra were dominated by DNA signals. DAPI-negative Gt-ABs carried only DNA.

DNA is one of the main components of Gt-ABs, like metazoan ABs[4,7,41,59]. The intensity and frequency of DNA peaks increased as the culture aged, suggesting that in the death phase, DNA is proportionally even more predominant in Gt-ABs (Table 1, Supplementary Data 1, Fig. 6). However, the intactness of the DNA is difficult to determine from the RS signature. The high intensity band present at 1584 cm⁻¹ (50–75% correlation), assigned to guanine and adenine vibrations, might indicate DNA

fragmentation[34]. While DNA fragmentation takes place during apoptosis, intact DNA could also be packed into ABs[41] potentially having a functional role in intercellular communication[41,45,59]. This feature is poorly understood in metazoan ABs. Stretching vibrations of the phosphodiester bonds (at 794, 823, 1073, 1079, 1091, 1094 and 1671 cm⁻¹)[26,29–32], observed almost exclusively in Gt-ABs harvested in the stationary phase, indicates DNA integrity during this stage.

Proteins exhibit the other major component detected in Gt-ABs. Proteins are also major constituents of metazoan ABs, either degraded[41] or intact[1,4,41]. The number of peaks assigned to proteins in Gt-ABs was similar in samples from the stationary or death phase, but the intensity of the peaks increased in samples harvested during the death phase (highest proportion of proteins; Fig. 6). The peak assignments mostly corresponded to individual amino acid signatures, predominantly phenylalanine and tyrosine, indicative of protein degradation (Table 1, Supplementary Data 1)[34,60]; however, amide I vibrations associated with different protein conformations (α-helix, β-sheet, coils and turns at 1641, 1665, 1686, 1691, 1697 cm⁻¹)[30–32] were present in Gt-ABs harvested in the stationary phase. Thus, a certain degree of protein structure appears to be conserved in Gt-ABs until protein degradation takes over during death.

The lipid composition of ABs has been found to differ from other extracellular vesicles and is characterized by phosphatidylserine in most animal cell types[4,41,59] (Supplementary Data 2). The externalization of phosphatidylserine to the outside of the plasma membrane serves as an "eat me" signal in metazoan PCD allowing neighboring cells to rapidly phagocytose dying cells, usually before complete self-dismantling[36,61]. Yeast has been shown to display phosphatidylserine externalization during cell death induced by oxidative stress[11]. In *G. theta* samples, however, peaks at 586, 588 and 593 cm⁻¹ [26–28,30–32] have been assigned to phosphatidylinositol and were only present in Gt-ABs in the stationary phase. We hypothesize that phosphatidylinositol

could be externalized in *G. theta* cells and be maintained in Gt-ABs. Phosphatidylinositol might be involved in death signaling, analogous to phosphatidylserine in ABs. Other lipid-specific Raman bands in Gt-ABs originated from phospholipid and cholesterol moieties[26,30–32], the main components of organelles and cell membranes. Fragments of these lipids most likely got packed into Gt-ABs (Fig. 2, Supplementary Fig. 1) as has been described for metazoan ABs[41].

In contrast to the biochemical composition of metazoan ABs, we observed carbohydrates in Gt-ABs, albeit only in DAPI-positive samples from the stationary phase. Starch is the predominant form of energy storage in phytoplankton and has been previously quantified by RS in green algae[28,54]. Starch globules were abundant and predominant in *G. theta* cells[24], and even apoptotic cells still contained starch reserves (Fig. 2), which were available to be packed into Gt-ABs. Accordingly, we detected peaks associated to amylose and amylopectin, the structural units of starch, at 400–407 and 471, 479 cm$^{-1}$. Peaks assigned to glycogen, the energy storage molecule of metazoans, surprisingly, were also found in Gt-ABs at 486 and 498 cm$^{-1}$.

The number of Gt-ABs increased with the age of the culture. FACS analyses indicated higher abundance of large Gt-ABs in the early stationary phase, while small Gt-ABs were more abundant during the death phase of the culture. Large Gt-ABs, therefore, might be degrading over time, reducing in size. We did not observe phagocytosis in *G. theta* cultures, Gt-ABs instead remained in the extracellular space until the complete demise of the culture. In contrast, metazoan ABs are quickly removed by phagocytic cells before triggering an immunological or inflammatory response to neighboring cells[41,45]. The contents of Gt-ABs seem to depend on their cells of origin, as reported previously for ABs[4]. In the stationary growth phase *G. theta* cultures contain a mixture of healthy, apoptotic, and dead cells (Fig. 1). Populations of apoptotic cells in this phase are expected to maintain their integrity for longer time, due to the nutrients released by fragmented dead cells. Gt-ABs are consequently packed with a more complex mixture of biomolecules, resulting in more complex Raman spectra for Gt-ABs harvested in stationary phase. This complexity is also reflected in the high number of peaks with ambiguous assignments (DNA + proteins, DNA + lipids, DNA/RNA + proteins, DNA/RNA + lipids, proteins + lipids, lipids + carbohydrates; Fig. 6). On the other hand, Gt-ABs harvested in the death phase are expected to be old ABs free-floating in the extracellular space for a long period of time. They might have decreased in size due to degradation of their inner components. Additionally, more dead cells are expected in the culture during the death phase than during the stationary phase, further simplifying the chemical composition of Gt-ABs (i.e., lower variance due to more uniformly degraded states of their cells of origin), and thereby their Raman spectra (Fig. 6).

The specific function of apoptosis and ABs in phytoplankton remains enigmatic, but their similarities to the metazoan apoptotic morpho-phenotype and ABs are striking: (1) Similar to metazoan cells, *G. theta* cells also display membrane blebs; (2) Gt-ABs exist as small membrane (and periplast) vesicles in the extracellular space; (3) Gt-ABs contain genetic material, such as DNA and RNA; (4) Gt-ABs display spectral signatures matching the major known constituents of metazoan ABs; (5) Based on RS, Gt-ABs carry fragments of the cell nucleus, chromatin remnants, other organelles, cytosol portions, degraded and intact proteins, and potentially energy storage carbohydrates typical of metazoan cells (glycogen).

The contents of metazoan ABs (DNA, microRNAs, proteins and lipids) have been found to mediate communication between cells[4,45]. Therefore, the in situ production of ABs in *G. theta* might be a signal to other cells, e.g., the increase in number of ABs is an alert signal of unfavorable physiological growth conditions in the natural environment, triggering cells to migrate to better conditions. *G. theta* cells are motile and contain accessory organelles, ejectisomes, that aid this process when subjected to a threat[23,24].

In an orchestrated process, *G. theta* might exhibit a type of intercellular communication mediated by extracellular vesicles or ABs[4,45] in natural environments within diverse phytoplankton populations. Phytoplankton blooms have been suggested to exhibit multicellular-like behavior mediated by chemical signals packed within extracellular vesicles potentially facilitating targeted exchange of information[25].

Apoptosis should be strictly defined by morphological hallmarks of cell disassembly, more specifically by the production of ABs[8,59], as seen in *G. theta*. Changes in size and shape of the cells, ultrastructural changes in the nucleus as well as the action of caspases are typical features of PCD but not restricted to apoptosis: they have been observed also in other non-apoptotic pathways of PCD. By determining the cellular and subcellular morphologies in aging *G. theta* cells we were the first to observe ABs, and thus apoptosis in phytoplankton. We have yet to discover the common ancestor of PCD, a missing link between uni- and multicellularity, and to determine if metacaspases (homologs of caspases) or other (still unknown) enzymes are the definite mediators of this process in unicellular organisms. It appears that death can't tell us apart: cryptophyte algal cells experience apoptotic death like animal cells in physiological conditions. Our discovery of AB production in *G. theta* sheds light into the process of PCD and impacts our understanding of the dynamics of cultures in the laboratory, the industry and in natural ecosystems.

## Methods

### *Guillardia theta* cultures

*Guillardia theta* (*G. theta*) CCMP2712 was obtained from the Bigelow National Center for Marine Algae and Microbiota. 1 L liquid cultures of *G. theta* were grown in Fernbach culture flasks in h/2 medium[62] (with a final concentration of 0.5 mM NH$_4$Cl) at 20 °C, constant shaking (50 rpm) and white light (30 μmol photons m$^{-2}$ s$^{-1}$) in a 12 h: 12 h, light: dark cycle[49]. Experiments were conducted in an AlgaeTron 230 light incubator (Photon Systems Instruments (PSI), Drasov, Czech Republic) using 4 biological replicates with a duration of 46 days in total, to assess apoptosis in standard growth conditions. Sampling took place on the onset of stationary phase (day 11), mid- and late stationary phase (days 17 and 24) and mid- and late death phase (days 28 and 42).

### Photosynthetic efficiency of *G. theta* cultures

Photosynthetic efficiency of the *G. theta* cultures was measured using the AquaPen-C AP-C 10 (PSI, Drasov, Czech Republic). Maximum quantum yield of Photosystem II (Fv/Fm) was taken every two days during exponential phase of growth and at different time intervals during stationary and death phases (1–5 days between measurements) due to the stability of the cell concentration during those stages.

### *G. theta* cell growth and quantification of apoptotic bodies (ABs)

Concentration of *G. theta* cells and *G. theta* ABs (Gt-ABs), and estimation of size during growth was performed with a Multisizer 3 Coulter counter (Beckman Coulter) and the Multisizer software (Version 3.53), following manufacturer's instructions. Cells and Gt-ABs were measured using an analytical volume of 100 μL with a 70 μm aperture. Two technical replicates were measured for each biological replicate ($n = 4$, Fig. 1).

### Culture harvesting, filtration and centrifugation for Gt-ABs enrichment

After the number of cells and Gt-ABs was determined, two types of samples were collected: whole cultures (mixture of cells and Gt-ABs) and filtered/centrifuged samples (enriched for Gt-ABs). At each sampling point, 2 mL samples of whole cultures were centrifuged at 7800 × *g* for 3 min and resuspended in artificial seawater for cell viability assays: SYTOX green, In Situ Cell Death Detection (TUNEL) and for Transmission Electron Microscopy (TEM).

For the Gt-ABs enrichment, whole cultures were subjected to the following filtration/centrifugation procedure: 10–15 mL of culture was vacuum filtered at very low pressure in 47 mm all glass filtration system (Millipore), using Munktell Filtrak™ Grade 1 F Qualitative High Purity Lab Filter Papers (Ahlstrom-Munksjo, pore size 5–6 μm). In the filtering flask, the recovered liquid (Gt-ABs and cells of small diameter), was mixed with 30–35 mL of Milli-Q® to wash the culture media from the samples. The filtration and washing procedure were repeated once more. All the volume was transferred to F50 tubes for centrifugation at 20 °C, $3220 \times g$ for 10 min to concentrate the Gt-ABs. Most of the supernatant was discarded but 2 mL were left in the F50 to resuspend the pellet. The sample was transferred to 2 mL Eppendorf tubes and centrifuged at $7800 \times g$ for 3 min, where a small white pellet with the Gt-ABs was collected after centrifugation. Small aliquots of the samples were examined under the light microscope (40X standard objective) between each centrifugation step to ensure the proper collection of the Gt-ABs. The enriched Gt-ABs samples were subsequently processed for TEM, Fluorescence-activated cell sorting (FACS) and RAMAN spectroscopy (RS).

### Cell death and apoptosis assessment by fluorescence microscopy

To assess cell viability, SYTOX Green (Invitrogen, Thermo Fischer Scientific) dead cell stain was added to the *G. theta* cultures at a final concentration of 1 μM and incubated in the dark for 15 min at room temperature (RT). The controls were prepared using a *G. theta* culture in exponential phase. The positive control was prepared by incubating the culture at 80 °C for 2 min and subsequently staining with SYTOX. The negative control consisted of an unstained culture. The experimental samples and the corresponding controls were imaged in a Fluorescence Microscope Leica DMi8a using a HC PL FLUOTAR L 40x/0.60 DRY objective and a DA/FI/TX filter to simultaneously capture chlorophyll autofluorescence and SYTOX green signal. Samples were stained, imaged, and counted to determine the percentage of dead cells in the culture at each collection day (Fig. 1b).

TUNEL assay (In situ Cell Death Detection Kit-Fluorescein, Roche) was used for detection and quantification of apoptosis. Whole culture samples in early death phase were centrifuged and concentrated to 1 mL for fixation with 1% paraformaldehyde (PFA) dissolved in artificial seawater. Samples were centrifuged at $7800 \times g$ for 3 min; the supernatant was discarded and the pellet resuspended in 1 mL of fixative. Using a Pelco Biowave pro+ (Ted Pella, Redding, CA) the samples went through an automated fixation process for 15 min that included two final rinses with PB buffer (0.1 M, pH = 7.4). Fixed cells were left at 4 °C until permeabilization and labeling. Fixed cells were washed twice with 500 μL of PBS buffer (pH = 7.5) and the buffer discarded to allow resuspension in 100 μL of ice cold, fresh permeabilization buffer (0.1% Triton-X 100 in 0.1% Sodium Citrate). After 2 min incubation on ice, cells were washed twice in 200 μL of PBS. To prepare the positive controls, fixed and permeabilized cells were incubated with TURBO DNase (6 U/mL) (Invitrogen, Thermo Fischer Scientific) for 30 min at 37 °C and subsequently inactivated and washed with PBS. The positive control and the experimental samples were resuspended in TUNEL reaction mix, which consisted of 50 μL of Enzyme solution added to 450 μL of Label solution. Negative controls consisted of only labelling solution. Controls and experimental samples were incubated for 37 °C in a humidified atmosphere in the dark and washed twice in PBS after incubation. All steps were carried out as per manufacturer's instructions. Samples and controls were directly analyzed in a Confocal Microscope Leica SP8 FALCON equipped with an HyD SMD1 detector, using an objective HC PL APO CS2 63x/1.40 oil and Fluorescence Lifetime Imaging Microscopy (FLIM). FLIM was used to differentiate between the fluorescein and chlorophyll autofluorescence signals (Fig. 1b).

### Gene expression analysis

**RNA extraction.** Total RNA was extracted using Invitrogen RNAqueous™ Total RNA Isolation Kit (part # AM1912, Thermo Fisher Scientific) with the following modifications: the frozen biomass pellets were mixed with 0.5 mL of Lysis/Binding Buffer (RNAqueous kit) and transferred to bead tubes type A (part # 740786.50, Macherey-Nagel). To perform tissue homogenization and lysis, the bead tubes were placed in the Bead beater (MM 400, Retsch) for 2 min at a frequency of 30 Hz and subsequently vortexed at 4 °C in the Vortex genie 2 (Scientific Industries) for 30 min at maximum speed (2850 rpm). The lysate that contained the RNA was centrifuged at 4 °C and $13,000 \times g$ for 10 min, then the supernatant was recovered in new 1.5 mL Eppendorf tubes and lysis/binding buffer was added to a final volume of 1 mL. Followed by another round of centrifugation (see conditions above), the clarified lysates were transferred to new 2 mL Eppendorf tubes and RNA isolation proceeded as per manufacturer's instructions. After RNA elution, quantity and quality were determined using a NanoDrop 2000C UV–Visible spectrophotometer (Thermo Fischer Scientific) and 2100 Bioanalyzer (Agilent Technologies).

**Quantitative reverse-transcription polymerase chain reaction (RT-qPCR).** Metacaspase gene expression was analyzed by RT-qPCR during exponential, stationary and death phases of the culture. RNA samples were treated with rDNaseI (Thermo Fisher Scientific) prior to cDNA synthesis with iScriptTM cDNA synthesis kit (Bio-Rad) using 100 ng/μL of template RNA and a reaction volume of 20 μL following manufacturer's instructions. RT-qPCR was conducted in a BioRad CFX 96 instrument, using SsoAdvanced™ Universal SYBR® Green Supermix (Bio-Rad). The $2^{-\Delta\Delta CT}$ method[63] was used for the analysis of the fold change in expression of the metacaspases GtMCA-I and GtMCA-III (formerly known as GtMC1 and GtMC2[49], respectively) during growth. Normalization was conducted with two reference genes, beta tubulin and dynein. The primer sequences used for RT-qPCR are listed in Supplementary Table 2. Expression levels are represented as relative to the gene expression at day 4 of growth, for each experimental condition. Data is expressed as the mean ± SEM. Two technical replicates were measured for each biological replicate ($n = 3$).

### Pharmacological induction of apoptosis

Staurosporine (STS) at a final concentration of 5 μM or Carbonyl Cyanide m-Chlorophenylhydrazone (CCCP) at a final concentration of 49 μM were added to *G. theta* cultures in exponential growth phase ($1 \times 10^6$ cells/mL), with DMSO as vehicle. The number of Gt-ABs and cells were measured every 2 h in a Multisizer 3 Coulter counter (Beckman Coulter), as previously described. Culture with 0.115% DMSO was used as a control. Two technical replicates were measured for each biological replicate ($n = 3$). Apoptotic cell morphology was also monitored with light transmitted-DIC (TL-DIC) microscopy for 4 h after STS and CCCP addition using a HC PL APO 63x/1.30 glycerin objective (DMi8 Inverted Microscope, Leica Microsystems). As comparison, the formation of Gt-ABs was monitored in an 8-week-old *G. theta* culture (death phase) (Supplementary Fig. 2).

### Ultrastructure analysis by Transmission electron microscopy (TEM)

Three types of samples were processed for TEM: whole cultures to track the ultrastructural changes in the cell population during aging and apoptosis, filtered samples enriched with Gt-ABs to capture their detailed ultrastructure, and purified sorted Gt-ABs (as described below). All types of samples were chemically fixed. Cells were fixed with 2.5% glutaraldehyde (GA; TAAB Laboratories, Aldermaston, England) and Gt-ABs were fixed with 1% PFA dissolved in artificial seawater. All types of samples were centrifuged at $7800 \times g$ for 3 min, the supernatant was discarded, and samples were resuspended in 1 mL of

fixative. Automated fixation proceeded as described in the previous section (TUNEL methods). During and after fixation, cells were centrifuged at $500 \times g$ for 2 min and Gt-ABs at $3000 \times g$ for 3 min. Rinses were performed with PB buffer (0.1 M, pH = 7.4) and samples were refrigerated at 4 °C until further processing.

Samples were further post-fixed for 2 h in 1% aqueous osmium tetroxide and embedded in 2% ultra-low melt agarose (Sigma-Aldrich) to reduce sample loss during processing. They were further dehydrated in ethanol and finally embedded in Spurr's resin (TAAB Laboratories, Aldermaston, England). All steps were performed using the Pelco Biowave pro +. Ultrathin sections (70 nm) cut with a Leica EM FC7 ultramicrotome (Leica Microsystems Inc.) were picked up on copper grids and post stained with 5% aqueous uranyl acetate and Reynolds lead citrate. Grids were examined with Talos L120C (FEI, Eindhoven, The Netherlands) operating at 120 kV. Micrographs were acquired with a Ceta 16 M CCD camera (FEI, Eindhoven, The Netherlands) using Velox software v2.14.2.40 (Fig. 2, Supplementary Fig. 1).

### Fluorescence-activated cell sorting (FACS) for Gt-ABs isolation

A combination of low-pressure filtration and centrifugation to enrich Gt-ABs in the samples was employed, followed by flow cytometry for their isolation and purification based on known biological characteristics. Although FACS has been shown to be sufficient to purify ABs up to 99%[59], we avoided technical problems during sorting by reducing the complexity of the samples with the enrichment steps.

Filtered samples enriched with Gt-ABs were fixed with 1% PFA and permeabilized (as described in TUNEL methods). Following permeabilization, samples were stained with DAPI (4′,6-diamidino-2-phenylindole, 50 μg/mL), incubated for 30 min at RT in the dark and washed twice with PBS. Pellets were resuspended in 70% FACS Flow buffer (BD Bioscience, San Jose, CA, USA) and filtered through 40 μm Flowmi Cell strainer (SP Bel-Art, Wayne, NJ, USA) just before analysis and sorting using a BD FACS Aria III flow cytometer equipped with four lasers: violet (405 nm), blue (488 nm), yellow-green (561 nm) and red (633 nm) lasers (BD Biosciences, San Jose, CA, USA). BD FACSDiva software version 8.0.3 was used for handling of the cytometer and data analysis. 70% FACS Flow was used as a sheath fluid. The Gt-ABs suspension was loaded in the cell sorter (4 °C, mild agitation at 100 rpm) and forced through the cuvette in a single-file stream. Light scatter information was collected at the sample interrogation point, where the laser lights intercepted the stream. The stream entered a 70 μm nozzle tip, where the drop drive broke the stream into the droplets for sorting according to the gate design.

Detection and recording of the Gt-ABs was done based on the sample's optical properties (forward scatter (FSC) and side scatter (SSC))[64]. The FSC was initially filtered through a 1.0 neutral density filter and then perceived by a photodiode detector with a 488/10 nm bandpass filter. Functional analysis was performed via chlorophyll and general auto-fluorescence as well as DAPI fluorescence excited by blue and yellow-green laser and violet laser, respectively. Autofluorescence emission spectra were perceived by 695/40 filter with 655 long pass (LP) mirror and 780/60 filter with 735 LP mirror while DAPI fluorescence was detected in intervals 450/40 nm and 610/20 nm with 595 nm LP mirror. The relevant size estimation of the suspension particles analyzed was facilitated by calibration beads with known diameters (2, 3 and 6 μm; BD biosciences). Purification of Gt-ABs from *G. theta* cells (size 5–14 μm) remaining in the enriched samples was possible due to their relative size difference and chlorophyll content.

A sequential gating strategy led to the selection of distinct Gt-ABs subpopulations (Fig. 3a(i)). Firstly, the identified population of *G. theta* cells with chlorophyll autofluorescence emission and relative size larger than 5 μm was excluded from sorting (Fig. 3a(ii), (iii)). Gt-ABs were then identified in the subsequent population based on their relative size (2–5 μm; Fig. 3a(ii)). Within this subpopulation of Gt-ABs, DAPI-

positive and -negative bodies were separately selected after comparison with the respective population of DAPI negative Gt-ABs in the unstained control samples (Fig. 3a(iv)–(vi)). Four subpopulations were identified and sorted: small DAPI-positive Gt-ABs (relative size c.a 2 μm), small DAPI-negative Gt-ABs, large DAPI-positive Gt-ABs (relative size c.a 3–5 μm) and large DAPI-negative Gt-ABs. The subpopulations of Gt-ABs were sorted and collected in round bottom polypropylene tubes in PBS buffer and kept at 4 °C during and post-sorting until processed for Raman spectroscopy (RS). Sampling points for stationary phase (days 17 and 24) and death phase (days 28 and 42), were sorted separately and then pooled together post-sorting for RS analysis. The FACS analysis provided the trends in the concentration of Gt-ABs (Fig. 3c, Supplementary Table 1) to complement the RS analysis of Gt-ABs during stationary and death phases.

### Confocal microscopy for Gt-ABs identity confirmation

20 μL aliquots of each of the sorted Gt-AB subpopulations identified by FACS were deposited in sterile confocal dishes (No. 100350, SPL Life Sciences) and imaged using a Zeiss LSM 880 inverted fast Airyscan confocal microscope (Carl Zeiss AG, Oberkochen, Germany). Images were taken with a C-Apochromat 40x/1.20 W Korr FCS M27 objective (Fig. 3b) and a Apochromat 63x/1.4Oil DIC objective (Fig. 3b, lower panel, DAPI (-) in stationary phase only), using one fluorescence track. The track configuration included the excitation of DAPI (405 nm violet laser) with emission detected between 410–585 nm for the AB and chlorophyll excitation (488 nm blue laser) with emission detection between 629–735 nm to check for the presence of *G. theta* cells by means of their chlorophyll autofluorescence (Fig. 3b).

### Confocal Raman spectroscopy (RS)

Sorted subpopulations of Gt-ABs were centrifuged at $7800 \times g$ for 5 min, concentrated and washed (3x) in Milli-Q® water to avoid any background signal from the PBS buffer. 5 μL droplets of Gt-ABs were deposited onto a square gold mirror (30 × 30 mm, OptoSigma Europe) and allowed to dry overnight, which also allowed the decay of DAPI signal completely. Imaging and dispersive Raman spectra were collected by a Renishaw Qontor spectrometer (using a 405 nm solid state laser, calibrated against the built-in Si standard). Spectra were collected in Stream HR static mode using 10 % (of total 40 mW) power and 0.2 s exposure time. A 2400 lines/mm grating was used, resulting in ca. $1\,cm^{-1}$ spectral resolution. A 100x standard lens in normal confocal mode was used in the spectral range 300–2500 $cm^{-1}$. Spectra were recorded, noise filtered and cosmic rays removed in Renishaw's WiRE software (version 5.6). Spectra were exported as tabulated ascii txt files and microscope white light images as .jpg files. At least 8 independent replicates of the Raman spectra were collected per sorted population per growth stage.

### Raman spectral analyses

Raman spectra were analyzed using a home-made, open source, MATLAB-based graphical user interface (MCR-ALS v6c Application, using MATLAB R2023a Update 5), based on the script by Felten et al.[65], freely available to download at the Vibrational Spectroscopy Core Facility at Umeå University (https://www.umu.se/en/research/infrastructure/visp/downloads/). In the MCR-ALS application, spectra were pre-processed by asymmetrical least squares baseline correction (lambda = 1000 and $p$ = 0.001), total area normalization and smoothing (Savitzky-Golay filter, polynomial order = 1, frame = 3), before running the multivariate curve resolution – alternating least squares (MCR–ALS) pipeline. The number of components were determined based on singular value decomposition, and in cases where clear cutoffs could not be established, several alternatives were tested. For MCR–ALS, only non-negativity constraints were applied (both for spectra and concentrations), and the convergence limit was set at 0.1.

Overlays of Raman maps on white light images provided visual correlation between the microscopic features and the molecular spectra, which was used for evaluating the reliability of the models and the quality of the spectra in the image maps (Fig. 4).

Principal component analysis (PCA) of the pre-processed spectra from images deemed to be of sufficient quality ($n = 99$) was performed in SIMCA 17 (Sartorius) to reduce the dimensionality of the data, investigate major trends, and find potential outliers (Supplementary Fig. 3). Following unsupervised PCA, supervised classification models were constructed via Orthogonal Projections to Latent Structures–Discriminant Analysis (OPLS–DA), to identify variation between and within Gt-ABs subpopulations in relation to the growth stage of the culture (Supplementary Fig. 4, Fig. 5). Data were centered, and the loadings shown are correlation-scaled. Default cross-validation settings (with 7 cross-validation groups) were used, and OPLS–DA models were restricted to a maximum of $1 + 2$ components (predictive + orthogonal) to avoid overfitting and facilitate direct comparisons between models. Only OPLS–DA models with the highest Q2 scores were considered for peak assignment and interpretation. Since this is the first report of RS applied to ABs, peak assignment was performed based on the specific molecular vibrations reported in literature and databases and compared to assignments of pure biological molecules, biological tissues, cells, and other types of extracellular vesicles. Assignment was determined by the occurrence of the exact Raman signature reported or with an accepted shift in the vibrational modes ($\pm 1$–$10\ cm^{-1}$) due to experimental factors, as well as chemical and biological variations.

## Statistical analyses

Statistical analysis was performed with GraphPad Prism (version 10.1.2; GraphPad Software Inc.) and the significance level was set at $\alpha = 0.05$ for all the analyses. Statistical tests used for data analysis and exact $P$- values are included in the figure legends and the Source data file. For the growth of the *G. theta* cell population and production of ABs in standard conditions, data are presented as mean values +/− standard errors of mean (SEM) of biologically independent samples ($n = 4$) and two technical replicates per sample. The differences between the means of the counts of Gt-ABs between the growth phases was determined by Brown-Forsythe ($F_{(2,\ 94.86)} = 126.7$), Welch's ANOVA ($F_{(2,\ 86.11)} = 203.7$) and Dunnett's T3 multiple comparisons test ($t = 13.62$, $df = 90.67$; $t = 17.63$, $df = 61$; $t = 7.87$, $df = 78.87$). For the expression of the metacaspases genes, data are presented as mean values +/− SEM of biologically independent samples ($n = 3$) and two technical replicates per sample. The statistical analysis of the differences between the means of the metacaspases gene expression (fold change) between the growth phases was performed separately for each gene. For GtMCA-I, Brown-Forsythe ($F_{(2,\ 30.60)} = 16.04$), Welch's ANOVA ($F_{(2,\ 17.74)} = 28.93$) and Dunnett's T3 multiple comparisons test ($t = 0.001278$, $df = 26.25$; $t = 7.302$, $df = 11.97$; $t = 5.548$, $df = 20.63$). For GtMCA-III, Brown-Forsythe ($F_{(2,\ 29.64)} = 10.38$), Welch's ANOVA ($F_{(2,\ 19.99)} = 24.31$) and Dunnett's T3 multiple comparisons test ($t = 0.02128$, $df = 90.29.95$; $t = 6.568$, $df = 14.61$; $t = 4.619$, $df = 22$). The significance of the pharmacological induction of apoptosis was analyzed with a two-way ANOVA (interaction, $F_{(6,24)} = 88.51$; row factor, $F_{(3,24)} = 234.8$; column factor, $F_{(2,24)} = 210.4$) followed by Dunnett's multiple comparison test. FACS data are presented as mean values +/− SEM of biologically independent samples ($n = 4$). The statistical significance analysis for the percentages of Gt-ABs subpopulations sorted by FACS were performed by two-way ANOVA (interaction $F_{(1,28)} = 42.71$, row factor $F_{(1,28)} = 0.8665$, column factor $F_{(1,28)} = 0.7644$) and multiple unpaired t-tests (small Gt-ABs, $t = 6.942$ $df = 14$ and large Gt-ABs, $t = 3.347$ $df = 14$). The number of events recorded during FACS were analyzed by growth phase using two-way ANOVA. The sources of variation were the interaction, the row and the column factors for small Gt-ABs ($F_{(1,28)} = 7.87$, $F_{(1,28)} = 219.1$,

$F_{(1,28)} = 8.51$, respectively) and large Gt-ABs ($F_{(1,28)} = 61.55$, $F_{(1,28)} = 272.1$, $F_{(1,28)} = 62.63$, respectively).

## Figures

All the figures were created using Affinity Publisher 2 (version 2.4.0.2301; Affinity software).

## Reporting summary

Further information on research design is available in the Nature Portfolio Reporting Summary linked to this article.

## Data availability

All the processed Raman spectra used in the paper are included in the Source Data file in a standard tabular format. The raw spectra as recorded on the instrument are available upon request to the corresponding authors. Source data are provided with this paper.

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

## Acknowledgements

The authors acknowledge the technical assistance of the facilities at the Chemical Biological Center (KBC), Umeå University: The Umeå Center for Electron Microscopy and the Biochemical Imaging Center - part of the National Microscopy Infrastructure (VR-RFI 2019-00217), the Vibrational Spectroscopy Core Facility and the Umeå Plant Science Center Microscopy Facility. We are grateful for the financial support by the Swedish Research Council (VR grant 2019-04472 to C.F.), the Sven och Lily Lawskis fond for the post-doctoral fellowship to L.C. and Umeå University. K.L., I.A. and V.S. were supported by grants from the Knut and Alice Wallenberg Foundation (KAW 2016.0352 and KAW 2020.0240) and the Swedish Research Council (VR grant 2021-04938). VS was supported by Kempestiftelserna (SMK21-0041).

## Author contributions

L.C. and C.F. conceptualized the study. L.C. and G.A.V. performed growth experiments, viability assays, TEM sample preparation, imaging and analysis, gene expression analysis. L.C. and B.J.W. performed fluorescence microscopy experiments. L.C., V.S., and I.A. developed the FACS methods. V.S. performed the FACS experiments. L.C. analyzed the FACS data. I.A. and K.L. supervised the FACS experiments. K.L. provided funding for FACS. L.C. and A.G. developed the methods and analyzed the Raman spectroscopy data. G.A.V. performed and analyzed the pharmacological induction of apoptosis. C.F. supervised the project and was responsible for funding acquisition. L.C. and C.F. wrote the manuscript with contributions from all authors.

## Funding

## Competing interests

The authors declare no competing interests.
