## [Transparent Peer Review file · Nature Communications]

Apoptotic Bodies in Phytoplankton Suggest Evolutionary Conservation of Cell Death Mechanisms

Corresponding Author: Professor Christiane Funk

Version 0:

Reviewer comments:

Reviewer #1

(Remarks to the Author)

In the paper "Death cannot tell us apart: Apoptotic bodies in phytoplankton", Corredor, L., et al, report the presence of apoptotic bodies, hallmark of apoptosis, in the cryptophyte *Guillardia theta* - an observation previously undescribed in non-metazoan single-celled eukaryotes. This raises important questions on the origin of Programmed Cell Death, in context of the origins of multicellularity. With the application of multimodal imaging techniques – confocal microscopy, Raman Spectroscopy and 2D Electron Microscopy, the authors have presented extensive cellular and subcellular support for their findings.

However, the manuscript needs considerable improvement in the quality of presentation (please see below for comments) and clarification on certain conclusions presented.

General Remarks

- Do metazoan apoptotic bodies (here termed as ApoBDs) need to be necessarily differentiated in terminology from Gt-ABs? The use of different terms such as apoptotic bodies (ABs), ApoBDs, algal bodies can be rather confusing.
- It should be clarified in the abstract itself that these apoptotic bodies are extracellular, to avoid confusion with autophagosomes which can also be membrane enclosed vesicles.
- All the figures need to be of high resolution, with legible labels.

Comments on Results and Figures

- Figure 1 A

- Unit for X axis is missing.
- There seems to be no account of correlation between the number of cells and Abs between the L and S phase. Why so? The increase in AB numbers seen at the start of S phase, indicating a high cell death rate but compensated by concomitantly dividing cells. The curves are mirrored from mid S phase to end of D, indicating division has either stopped or decreased significantly, and cells are largely dying. Hence, suggesting a negative correlation between number of cells and number of ABs. It would also be interesting to validate the above with a BrdU assay for net cell counts?
- What explains the steady state between Day 24 and 29? Both the number of cells and ABs plateau?

- Figure 1B

- Chlorophyll autofluorescence is shown in red in the SYTOX assay, while in blue in the TUNEL assay. Consistent LUTs would be easier to follow.
- Moreover, while the scale bars correspond to 20 μm in both the panels, cells look notably different in size (SYTOX being

smaller than TUNEL).

- In the TUNEL panel, blue and green are barely distinguishable from each other.

- Figure 2

- The resolution of the micrographs is not sufficient to visualize or comment on the ultrastructural hallmarks described in the results.
- The dynamic nature of the blebs is hard to extrapolate solely from TEM data. What do you think of the electron lucent vesicles in the cytoplasm as in B and D.
- The authors suggest that the exclusion of ABs leaves empty cytosolic space (C and D) but could this be from mechanical damage during sample preparation?
- Was TEM done for sorted populations of whole cells and ABs? From the micrographs, it appears a pellet with a mix of cells and free-floating ABs was sectioned. As such, it is difficult to show if the ABs have indeed no connection with the whole cell and are 'free' or simply a matter of plane of sectioning.

- Figure 3

- Other than what is quantified in C, panel B doesn't seem to be adding any extra information. While the authors remark different objectives were used for enhanced details, this cannot be visualized in the images.

- Figure 4

- Legend is swapped for 4A and B. Confocal RS images are presented in A rather than B as stated in the legend. Moreover, labels on the confocal images are barely legible and better image quality is needed.

- With regards to the FACS analysis of the Gt-ABs, it would be good to state what the % mean. It should be stated more explicitly that it is the relative proportion within the ABs population. In line with this, the authors should rephrase their results, as they do not tell anything about the total "number" of ABs, just the proportion (line 150 – "the number of large Gt-ABs significantly decreased").

- Raman spectroscopy analysis, although beyond my domain of expertise, appears to be comprehensive, well performed and provides important information for the scope of the study.

Open questions

- The authors describe the exclusion of ABs as a progressive process, with early stages of intracellular ABs. Could it be possible that these are simply food vacuoles, autophagosomes? Are *G. theta* known to be mixotrophic?
- It would be interesting to analyse the content of ABs more extensively, in terms of volume density and compare between the whole cells and ABs. As more and more organelles are packed into ABs, do the whole cells show a decline in their organellar content?
- An interesting follow up experiment for the future might be to supplement fresh cultures with isolated ABs to see if these indeed aid in the health of a culture.
- Is there a general consensus to define apoptotic events strictly by morphology and ultrastructural hall markers? For the future, it would be worthwhile to complement morphology and subcellular studies with molecular characterization of the process.

Reviewer #2

(Remarks to the Author)

Reviewer #3

(Remarks to the Author)

The manuscript "Death cannot tell us apart: Apoptotic bodies in phytoplankton" presents groundbreaking research on the discovery of apoptotic bodies (ABs) in the unicellular alga *Guillardia theta*, a non-metazoan eukaryote. This discovery has significant implications for our understanding of programmed cell death across different organisms. This is the first time apoptotic body production has been observed in phytoplankton. The ABs produced by *G. theta* have a similar composition to metazoan ApoBDs, containing degraded and intact DNA and proteins, lipids, energy storage carbohydrates, and fragments of organelles and cytosol. The function of AB production in *G. theta* remains unclear, but the authors think that ABs might act as extracellular signals in phytoplankton in natural environments. These new findings were presented well, and would contribute to the understanding of cell death mechanisms in phytoplankton and has implications for our knowledge of evolutionary biology and ecosystem dynamics. A lot of effort has been invested into this research. The methods fit the

objectives. I think that it is an interest and worthwhile topic to be published in STOTEN, but there are some problems to revise. A minor revision is required to make the paper more understandable.

The comments are summarized below:

- 1) Line 203-207, the authors think three of the Gt-AB subpopulations, showed obvious differences in their spectra originating from stationary or death phase, with higher intensities of peaks in the death phase, but there is higher peak intensity of large DAPI-negative Gt-AB in the stationary phase. How to explain it?
- 2) Line 366-369, how does the authors get to know there are single stranded or fragmented DNA in the Gt-Abs?
- 3) Line 371, what's the meaning of NA?
- 4) The authors mention the genes and proteins that regulate ABs production. Can they apply transcriptomic and proteomic analyses to study the functions of these genes and proteins in ABs?

Reviewer #4

(Remarks to the Author)

Corredor and colleagues investigate whether apoptotic bodies are generated by aged Cryptophytes, specifically *Guillardia theta*, a unicellular microalga, using Confocal Microscopy, Transmission Electron Microscopy (TEM), Fluorescence-Activated Cell Sorting-FACS, and Confocal Raman Spectroscopy (RS). The authors have done an excellent job in identifying released bodies and its content, providing well-documented evidence that represents an important advance in the field of cellular communication in unicellular organisms.

Demonstrating the release of apoptotic bodies would be a novel finding. However, to determine whether these structures are truly apoptotic bodies or extracellular vesicles that facilitating cell-to-cell communication in this aged organism, like others found in another algae or cyanobacteria (Paterna, et al. 2022, DOI:10.3389/fbioe.2022.836747, Biller et al. 2014, DOI:10.1126/science.124345), it is essential to follow the recommendations of the Nomenclature Committee on Cell Death (NCCD), cited in the manuscript, and, for example, those outlined by Aguilera et al., 2021,- DOI: 10.3389/fmicb.2021.631654 i.

Some of my major concerns in this regard:

The study of apoptotic bodies of Corredor and colleagues is supported by morphological and size-based evidence.

However, according to the NCCD guideline, biochemical, genetic, and pharmacological approaches should be integrated in order to robustly support the statement.

- The exploration of molecular markers or signalling pathways associated with apoptosis in *G. theta* could further substantiate these findings. Including such analyses would provide more robust evidence supporting the apoptotic origin of the observed bodies.
- The inclusion of pharmacological studies in the manuscript would significantly enhance the robustness of the findings. For instance, the use of apoptosis inducers such as staurosporine (STS) or caspase inhibitors could help elucidate whether caspases are involved in the formation of apoptotic bodies in *G. theta*. While most apoptosis inducers and inhibitors so far have been exclusively used on metazoans, some have been applied to the photosynthetic eukaryotic and prokaryotic organisms for demonstrating specificity of a particular type of cell death (Dixon et al 2012, 10.1016/j.cell.2012.03.042, Ramachandran et al 2023, DOI: 10.3390/cells12040553, Aguilera et al 2022, DOI: 10.1083/jcb.201911005, Carbó et al 2023, DOI: 10.1016/j.jhazmat.2023.130997

Line 318. Some assays to measure membrane potential and organelle functionality could be highly interesting and valuable for this study (eg., JC-1 staining)

Line 324-343. I found the paragraph between lines 324-343 somewhat contradictory. Firstly the authors list some important differences that exist between apoptotic cells in metazoans and in *G. theta*, but the conclusion state that “*G. theta*’s apoptotic morpho-phenotype is highly identical to metazoan apoptosis”. Please clarify.

Version 1:

Reviewer comments:

Reviewer #1

(Remarks to the Author)

I am generally satisfied with the new version of the manuscript, as the authors took good care at addressing most of the points that we have raised.

I still have some reserves and important points related to the EM work:

- major point: sorry to insist, but neither from the text, nor from the figure legend, it appears clear to me that the ABs are imaged from FACS sorted small particles or from whole cultures that include cells. If the latter, then I am afraid it is not possible to unambiguously identify free floating AB. A 2D plane in TEM can section and display a budding Abs without the showing cell it’s budding from.

In the legend, authors write “Extracellular Gt-ABs, free floating in solution, ..” and in the text “Extracellular, free-floating Gt-Abs ...” such sentences do not clearly state if those originate from a sorting of particles that would exclude the whole cells.

From these I would tend to interpret that EM was done only on cell cultures and not on sorted / filtered ABs.

Yet, looking at the methods section, it seems that TEM experiments were performed on both whole cultures and on purified

ABs. The results should clearly state which conditions the images are coming from.

- minor point: on figure 2, please be consistent in labelling the nuclei or nucleoli. "N" is use in the first panels and this not defined in the legend. I assume the refer to the nuclei there.

Reviewer #3

(Remarks to the Author)

The authors have made huge improvements to their manuscript. I can recommend it for publication.

Reviewer #4

(Remarks to the Author)

The authors have conducted an excellent and thorough revision in response to all of my previous questions and concerns. Their replies are clear, well-supported by new experiments and thoughtful discussion, and demonstrate the robustness of their findings. I particularly appreciate the inclusion of additional data and methodological clarifications, which have substantially improved the manuscript.

Based on these revisions, I believe the manuscript now meets the expected standards for publication in this journal. The conclusions are well supported by the data, the methodology, and sufficient detail is provided to ensure reproducibility. The results are relevant to the field, and the study offers valuable contributions to the existing literature.

Therefore, I recommend acceptance of the manuscript in its current form.

Version 2:

Reviewer comments:

Reviewer #1

(Remarks to the Author)

The answer to my concerns is clear and I am happy with the revised text.

Christiane Funk, Ph.D.
Professor in Biochemistry

Point-by-point response to the reviewers' comments

Reviewer #1 (Remarks to the Author):

In the paper "Death cannot tell us apart: Apoptotic bodies in phytoplankton", Corredor, L., et al, report the presence of apoptotic bodies, hallmark of apoptosis, in the cryptophyte *Guillardia theta* - an observation previously undescribed in non-metazoan single-celled eukaryotes. This raises important questions on the origin of Programmed Cell Death, in context of the origins of multicellularity. With the application of multimodal imaging techniques – confocal microscopy, Raman Spectroscopy and 2D Electron Microscopy, the authors have presented extensive cellular and subcellular support for their findings.

However, the manuscript needs considerable improvement in the quality of presentation (please see below for comments) and clarification on certain conclusions presented.

General Remarks

- Do metazoan apoptotic bodies (here termed as ApoBDs) need to be necessarily differentiated in terminology from Gt-ABs? The use of different terms such as apoptotic bodies (ABs), ApoBDs, algal bodies can be rather confusing.

We thank the reviewer for this comment to improve our manuscript. Even though metazoan apoptotic bodies are referred to as "ApoBDs" in some recent reviews and original publications, after performing a literature search on the nomenclature we concluded that the term "apoptotic bodies" or "ABs" are the most used. We agree with the reviewer and to avoid confusion with many similar terms, the acronym ApoBDs has been replaced for ABs throughout the text. However, we decided to continue to use the acronym Gt-ABs to enable the reader to easily distinguish between the newly detected ABs in phytoplankton and metazoan ABs. Further, in line 78, the term "algal bodies" has been replaced by the term "extracellular vesicles" (see also next comment).

- It should be clarified in the abstract itself that these apoptotic bodies are extracellular, to avoid confusion with autophagosomes which can also be membrane enclosed vesicles.

We thank the reviewer for this suggestion and added the word "extracellular" in line 22 and 25 of the abstract. We further have added Supplementary Data Table 2 to the manuscript, showing a comparison between the different types of vesicles produced by various organisms to further highlight the differences between ABs and other types of vesicles reported in literature. We refer to this table in the discussion (Lines 407-413), in the section "Ultrastructural and subcellular markers of apoptosis and AB production"; as well as in the section "Biochemical characterization of Gt-ABs, insights into ageing and death" (line 475).

- All the figures need to be of high resolution, with legible labels.

We agree with the reviewer; all the labels have been modified accordingly. The figures were (and are also in this resubmitted manuscript) embedded in the manuscript following the publisher's guidelines for submission. Graphs, charts and schematics will be provided in vector format once the manuscript is accepted, and figures will be produced in high resolution. To facilitate visualization of the figures for the reviewers, we additionally uploaded high resolution images for resubmission.

Comments on Results and Figures

- Figure 1 A
- Unit for X axis is missing.

We apologize for this mistake and now have added the unit.

- There seems to be no account of correlation between the number of cells and ABs between the L and S phase. Why so?_The increase in AB numbers seen at the start of S phase, indicating a high cell death rate but compensated by concomitantly dividing cells. The curves are mirrored from mid S phase to end of D, indicating division has either stopped or decreased significantly, and cells are largely dying. Hence, suggesting a negative correlation between number of cells and number of ABs. It would also be interesting to validate the above with a BrdU assay for net cell counts?

We agree with the reviewer. The G. theta cells follow a typical growth curve known for microalgae and many bacteria. In a fresh culture, algae cells are highly diluted and will need some days to adjust to the new environment (lag phase). Healthy algal cells in growth medium providing enough nutrients are then dividing with a specific doubling time, visible by the logarithmic phase of the growth curve indicating exponential (E) growth. In our hands G. theta has a doubling time of 72 hours. When nutrients become limited, growth (i.e. doubling of the cells) is stagnating (stationary phase) and some microalgal cells will start to die and release ABs. Without addition of fresh media/nutrients, also the remaining cells will then die (death phase). In the lag (L) and exponential (E) phase the number of Gt-ABs remains low, because healthy microalgal cells are dividing instead of dying. While towards the end of the stationary (S) phase and the death (D) phase cells are dying and releasing ABs instead of dividing. Our data show that the release of Gt-ABs does not support growth of the remaining few living cells (see also additional figure below, in your open questions section). Currently we do not know how many ABs are released from one G. theta cell.

Indeed, a BrdU assay sounds highly interesting, which has been used for diatoms and might work in cryptophytes. However, we first must establish the method at the FACS facility of Umeå University and then will have to measure during all growth phases (the culture grows for 40-50 days). Therefore, due to time limitations, we were not able to perform the experiment but will try to perform it in the future.

- What explains the steady state between Day 24 and 29? Both the number of cells and ABs plateau? Yes, but why? Reasons?

We thank the reviewer for this observation. We have no explanation for this plateau. For some reason during these days growth of the *G. theta* population stagnated, they neither died, releasing more ABs, nor did they double. Alternatively, at day 22 and day 29 the cell count for *G. theta* cells was slightly higher. Our hypothesis is that during different periods of the stationary phase (which in *G. theta* is very long) there are events of nutrient recycling. This means that the contents of cells that suffer complete demise are released into the medium and the population of cells still viable in the culture can uptake these nutrients and use them for cell doubling.

- Figure 1B

- Chlorophyll autofluorescence is shown in red in the SYTOX assay, while in blue in the TUNEL assay. Consistent LUTs would be easier to follow.

We have recolored the primary data in the SYTOX assay to a color-safe combination, the chlorophyll autofluorescence is shown in magenta and SYTOX green is presented as green fluorescence signal.

Although we agree with the reviewer's opinion about consistent LUT's, we consider that is important to show that the colours in the TUNEL experiment are different to the SYTOX stain to indicate the differences in the experimental approach. TUNEL fluorescein lifetime is shown as green fluorescence versus the chlorophyll autofluorescence lifetime (blue) in *G. theta* cells. For clarification, we added in Figure 1 the lifetime colour bar with the corresponding explanation in the figure legend and the result section (lines 109-117).

- Moreover, while the scale bars correspond to 20 μm in both the panels, cells look notably different in size (SYTOX being smaller than TUNEL).

We thank the reviewer for pointing out this error. The figures were produced and exported using Affinity software, the zoom used for the images was different in the panels but has been corrected to render a consistent representation of all the images. In addition, the images were taken with different objectives and microscopes. SYTOX assays were performed in a Fluorescence Microscope Leica DMI8a using a HC PL FLUOTAR L 40x/0.60 DRY objective and a DA/FI/TX filter to simultaneously capture chlorophyll autofluorescence and SYTOX green signal. In addition, we have corrected the scale bars for the SYTOX images, they correspond to 10 μm (also clarified in the figure legend).

TUNEL assay samples were analyzed in a Confocal Microscope Leica SP8 FALCON equipped with an HyD SMD1 detector, using an objective with better resolution (HC PL APO CS2 63x/1.40 oil) and Fluorescence Lifetime Imaging Microscopy (FLIM).

- In the TUNEL panel, blue and green are barely distinguishable from each other.

We thank the reviewer for this observation but as stated before, we consider that the colours should be retained since they provide information of the FLIM microscopy used for the TUNEL assay. Indeed, there is an overlap between the autofluorescence and the fluorescein fluorescence of the TUNEL assay, blue and green colour respectively. Using FLIM we distinguished the two signals, with autofluorescence displaying shorter lifetime (close to 0.65 ns) and fluorescein longer lifetime (close to 3.94 ns). The results from the FLIM analysis are now added to the manuscript (Lines 114-117). In addition, we believe that the distinction between blue and green in the high-resolution image will not be a problem for the reader (Please see our answer to the reviewer's comment concerning Figure 1B).

- Figure 2

- The resolution of the micrographs is not sufficient to visualize or comment on the ultrastructural hallmarks described in the results.

As mentioned above with this manuscript we will provide figures with higher resolution, their resolution still will be further improved after acceptance.

- The dynamic nature of the blebs is hard to extrapolate solely from TEM data. What do you think of the electron lucent vesicles in the cytoplasm as in B and D.

We thank the reviewer. To support our findings, we now have included the pharmacological induction of apoptosis (Extended Data Figure 2), which illustrates further the dynamic nature of the membrane blebs in a time lapse.

Regarding the electron translucent vesicles, we hypothesize that they are vesicles for intracellular trafficking that are affected by cell death processes and in consequence are empty and translucent in EM. In healthy cells they are electron dense in EM (this phenomenon has been reported in animal tissues). Nitrogen deficiency in Chlamydomonas reinhardtii leads to the formation of translucent vacuoles, similar to what is observed in G. theta cells in the death phase (<https://doi.org/10.1128/ec.00203-09>). Therefore, we can hypothesize that these vacuoles are connect to the nutrient depletion in the death phase G. theta culture.

- The authors suggest that the exclusion of ABs leaves empty cytosolic space (C and D) but could this be from mechanical damage during sample preparation?

We thank the reviewer for this comment; however, we are certain that we indeed observe empty cytosolic space. The same phenomenon has been observed in various experiments using different sample preparations. The sample preparation for TEM was carefully optimized throughout the course of several trials prior to the actual experiments. In the final experiments, ABs were embedded in 2% ultra-low melt agarose prior to post-fixation in aqueous osmium tetroxide, to precisely avoid mechanical damage during sample preparation.

Optimisation also included:

- ***Chemical fixation:*** we tested various published methods and incubation/washing times (with and without the microwave).
 - ***Reagents and their concentrations:*** Formaldehyde (1 to 4%) and glutaraldehyde (1 to 2.5%) in phosphate buffer and/or seawater.
- Was TEM done for sorted populations of whole cells and ABs? From the micrographs, it appears a pellet with a mix of cells and free-floating ABs was sectioned. As such, it is difficult to show if the ABs have indeed no connection with the whole cell and are 'free' or simply a matter of plane of sectioning.

We thank the reviewer for this comment. We performed TEM on non-sorted as well as on sorted populations. We would like to refer to Figure 2E and the Extended data Figure 1, where detached bodies completely isolated from the cells can be observed. The latter were images taken from an experiment with sorted bodies.

Both our data using FACS and confocal RAMAN spectroscopy proof that indeed Gt-ABs are entities separated from the cells. We would not have been able to sort the bodies if they were attached to the cell, the gating strategy was designed

to sort individual particles (not aggregates) and was based on complexity, size and lack of chlorophyll fluorescence.

Confocal RAMAN imaging also allowed visual confirmation of free-floating Gt-ABs sorted and completely separated from cells. Gt-ABs differ from *G. theta* cells in size and the lack of pigments as shown in the figures below.

G. theta cells:

Gt-ABs:

Figure A: Confocal RAMAN spectroscopy on *G. theta* cells (upper panel) and Gt-ABs (lower panel).

In addition, the Extended data Figure 1 (panels G and H) has been updated to further illustrate the differences between *G. theta* cells and bodies.

- Figure 3

- Other than what is quantified in C, panel B doesn't seem to be adding any extra information. While the authors remark different objectives were used for enhanced details, this cannot be visualized in the images.

Figure 3B was added to the manuscript to highlight A) the important information that Gt-ABs stay similar in appearance independent of the culture phase; B) to provide visual confirmation that indeed free Gt-ABs were sorted, and C) to provide visual confirmation of the subpopulations with and without DAPI signal. Images of the Gt-ABs were taken using two different objectives to provide a better overview. The figure legend has been updated accordingly and the scale

bar for the higher resolution objective (DAPI-negative in stationary phase) has been corrected in the image and the legend.

This is the first time that ABs in phytoplankton are reported, we therefore feel, we should provide as much visual proof of these vesicles as possible.

- Figure 4

• Legend is swapped for 4A and B. Confocal RS images are presented in A rather than B as stated in the legend. Moreover, labels on the confocal images are barely legible and better image quality is needed.

We very much thank the reviewer for this comment and apologize for the error. We now corrected A and B in the legend and, as explained earlier, will provide images with higher resolution. In addition, we removed the unnecessary labels on the confocal images to not distract the reader from the important information.

- With regards to the FACS analysis of the Gt-ABs, it would be good to state what the % mean. It should be stated more explicitly that it is the relative proportion within the ABs population. In line with this, the authors should rephrase their results, as they do not tell anything about the total “number” of ABs, just the proportion (line 150 – “the number of large Gt-ABs significantly decreased”).

We thank the reviewer for this correction. We now state explicitly in the paragraph that the “relative proportions” change (Line 184). We further clarify in the legend of Figure 3.

- Raman spectroscopy analysis, although beyond my domain of expertise, appears to be comprehensive, well performed and provides important information for the scope of the study.

We thank the reviewer for this comment. Our aim was to give a thorough biochemical composition of the Gt-ABs.

Open questions

- The authors describe the exclusion of ABs as a progressive process, with early stages of intracellular ABs. Could it be possible that these are simply food vacuoles, autophagosomes? Are *G. theta* known to be mixotrophic?

We thank the reviewer for this interesting observation. Our experiments were conducted in autotrophic growth conditions, in the growth medium of *G. theta* no carbohydrates are present, but we cannot exclude mixotrophy - carbohydrates could originate from dead cells. According to literature cryptophytes can only grow autotrophic. Phagotrophy has been reported for freshwater cryptophytes (*G. theta* is a seawater strain):

<https://academic.oup.com/plankt/article/47/1/fbae077/7989936>

To investigate the role of Gt-ABs in the culture, isolated Gt-ABs were added to *G. theta* cells in the exponential growth phase (Figure below). Before the addition of Gt-ABs the *G. theta* cells were transferred to minimal media (artificial sea water, ASW). ASW is the basis of h/2 medium, which is used to grow *G. theta*, but lacks vitamins, trace metals and nutrients. Gt-ABs were isolated from cultures in the death phase. The cultures were filtered using Munktell Filtrak™ Grade 1F

Qualitative High Purity Lab Filter Papers (Ahlstrom-Munksjo, pore size 5–6 μm), as described in the manuscript. The Gt-ABs and small cells (which were able to pass the filter, ca 4 μm), were centrifuged at 3220 x g for 10 min at 20 °C, washed twice and resuspended in ASW. To study if Gt-ABs would provide nutrients for *G. theta* cells, the cell number was monitored for 18 days. As a control, *G. theta* cells in the exponential phase were grown in ASW in the absence of Gt-ABs. Also, Gt-ABs (containing a few small cells) were grown in ASW as a control. The number of cells (5-14 μm) and Gt-ABs (2-5 μm) in the cultures were counted in a Multisizer 3 Coulter counter (Beckman Coulter).

After 18 days there was no significant difference observed in growth of the culture containing cells as well as Gt-ABs (A), the culture growing in the absence of Gt-ABs (just cells, B) or Gt-ABs (C). Gt-ABs appear to have no effect on culture growth. Furthermore, in the culture containing only Gt-ABs, a decline of Gt-ABs (both 2 μm and 3-5 μm populations) and the remnant small cells (5-14 μm) was observed. In cultures deprived of nutrients and vitamins, *G. theta* cells therefore are not able to use Gt-ABs as food vacuoles, which is leading to culture decline.

Figure B: Addition of Gt-ABs to a *G. theta* culture growing in minimal media. Further observations of *G. theta* cultures left in the incubator after the end of the experiments (1- 2 months), show complete demise of the culture, despite the high numbers of Gt-ABs counted in the last days of the ongoing experiment, confirming the previous observation.

- It would be interesting to analyse the content of ABs more extensively, in terms of volume density and compare between the whole cells and ABs.

We thank the reviewer for this comment; our current data are qualitative, but in future studies we will perform quantitative experiments. Size estimation was performed with the Multisizer 3 Coulter counter (Beckman Coulter), the size of Gt-ABs range from 2–5 μm, while *G. theta* cells range from 5–14 μm (graphs from the coulter can be provided, if necessary). Additionally, the sizes of cells and Gt-ABs were estimated using a BD FACS Aria III flow cytometer. Particle size was

analysed by calibration beads with known diameters (2, 3 and 6 µm). TEM images provide a visual confirmation of the size, we refer to Extended data Figure 1. Confocal RAMAN spectroscopy was used to extensively determine the biochemical content of the Gt-ABs. These images provide further visual confirmation of the differences between cells and Gt-ABs. We now added a Raman figure to Extended data Figure 1, for further confirmation of the differences between the G. theta cells and of the Gt-ABs.

The content of Gt-ABs in terms of volume density is more related to the next open question. We agree that it is an interesting aspect, and we have taken it into consideration. This work has opened new avenues of research for our lab and when we have access to new funding, many future experiments will be performed.

As more and more organelles are packed into ABs, do the whole cells show a decline in their organellar content?

We thank the reviewer for this interesting comment. Based on our TEM analyses (Figure 2) we agree with the reviewer's statement, however, we do not believe that whole organelles are packed into the ABs, rather remnants or shrunken non-functional organelles. Please keep in mind that the ABs do not contain any chlorophyll (arising from chloroplasts). Further studies will have to show the fate of the organelles in dying G. theta cells.

- An interesting follow up experiment for the future might to be to supplement fresh cultures with isolated ABs to see if these indeed aid in the health of a culture.

We thank the reviewer for this interesting approach. As shown above, we have performed such an experiment (Figure B in this response document).

- Is there a general consensus to define apoptotic events strictly by morphology and ultrastructural hall markers? For the future, it would be worthwhile to complement morphology and subcellular studies with molecular characterization of the process.

We thank the reviewer for the comment, and we agree. To provide a molecular characterization we have now added the gene expression data of the two metacaspases present in the genome of G. theta (Figure 1C). Metacaspases are structural relatives to the metazoan caspases, which are important players in metazoan PCD. Plants and algae do not contain caspases, but structural relatives called metacaspases, which are – based on the organisation of their catalytic p20 domain and regulatory p10 domain - subdivided into type I, II or III. G. theta contains genes coding for one type I and one type III metacaspase. We have also added data, where we present the pharmacological induction of PCD (Extended Data Figure 2).

In addition, we would like to point out that the TUNEL assay is considered a form of molecular characterization. It provides insights into the molecular mechanisms of cell death, focusing on DNA fragmentation, one of the hallmarks of apoptosis. By detecting, TUNEL assays help characterize the molecular state of the cells that are undergoing programmed cell death.

Reviewer #2 (Remarks to the Author):

We thank all reviewers for their time and effort to improve our manuscript.

Reviewer #3 (Remarks to the Author):

The manuscript "Death cannot tell us apart: Apoptotic bodies in phytoplankton" presents groundbreaking research on the discovery of apoptotic bodies (ABs) in the unicellular alga *Guillardia theta*, a non-metazoan eukaryote. This discovery has significant implications for our understanding of programmed cell death across different organisms. This is the first time apoptotic body production has been observed in phytoplankton. The ABs produced by *G. theta* have a similar composition to metazoan ApoBDs, containing degraded and intact DNA and proteins, lipids, energy storage carbohydrates, and fragments of organelles and cytosol. The function of AB production in *G. theta* remains unclear, but the authors think that ABs might act as extracellular signals in phytoplankton in natural environments. These new findings were presented well, and would contribute to the understanding of cell death mechanisms in phytoplankton and has implications for our knowledge of evolutionary biology and ecosystem dynamics. A lot of effort has been invested into this research. The methods fit the objectives. I think that it is an interesting and worthwhile topic to be published in STOTEN, but there are some problems to revise. A minor revision is required to make the paper more understandable.

The comments are summarized below:

1) Line 203-207, the authors think three of the Gt-AB subpopulations, showed obvious differences in their spectra originating from stationary or death phase, with higher intensities of peaks in the death phase, but there is higher peak intensity of large DAPI-negative Gt-AB in the stationary phase. How to explain it?

We thank the reviewer for bringing this aspect to our attention. The statement corresponding to lines 203-207 (now lines 243-248) is a qualitative observation from the average spectra and the peak intensities corresponding specifically to carbohydrates, nucleic acids, proteins and lipids (highlighted in Figure 4B). Indeed, this generalization is not correct, higher peak intensity of large DAPI-negative Gt-AB in the stationary phase can be observed. For this reason, the original statement and the discussion has been modified. In the stationary growth phase *G. theta* cultures contain a mixture of healthy, apoptotic, and dead cells (Figure 1). Populations of apoptotic cells in this phase are expected to maintain their integrity for longer time, due to the nutrients released by fragmented, dead cells. Gt-ABs are consequently packed with a more complex mixture of biomolecules, resulting in more complex Raman spectra for Gt-ABs harvested in stationary phase. This complexity is also reflected in the high number of peaks with ambiguous assignments (DNA + proteins, DNA + lipids, DNA/RNA + proteins, DNA/RNA + lipids, proteins + lipids, lipids + carbohydrates) (Figure 6).

2) Line 366-369, how does the authors get to know there are single stranded or fragmented DNA in the Gt-Abs?_

We thank the reviewer for this interesting reflection.

A) We detected apoptotic DNA fragmentation in nearly 50% of the cells in the culture during the death phase, confirmed by TUNEL experiments. The TUNEL reaction preferentially labels DNA strand breaks generated during apoptosis with high specificity. This allows discrimination of apoptosis from necrosis and from primary DNA strand breaks induced by cytostatic drugs or irradiation.

B) Whereas *G. theta* cells growing in the exponential phase displayed well defined nuclear membranes as well as chromatin localized in the nucleolus (Extended data Figure 1C,E), during apoptosis we observed chromatin condensation and its migration to the nuclear envelope (Figure 2A,C). DNA fragmentation and complete degradation became apparent with the loss of nuclei and/or the nucleolus. In metazoans, similar features are very well described and mark the early stages of apoptosis. They are also well documented in the chlorophyte alga *Dunaliella tertiolecta* during PCD induced by light deprivation. The morphological changes observed in the nucleus of *G. theta* cells during early apoptosis correlated well with the DNA degradation detected by TUNEL in the culture death phase.

C) The intactness of the DNA is difficult to determine from the Raman spectroscopy signatures; however, we found the high intensity band present at 1584 cm⁻¹ (50–75% correlation), assigned to guanine and adenine vibrations. Vibrations corresponding to nucleotides might indicate DNA fragmentation, in contrast to stretching vibrations of the phosphodiester bonds (at 794, 823, 1073, 1079, 1091, 1094 and 1671 cm⁻¹) that related to the structural integrity of the nucleic acids. The latter were observed almost exclusively in *Gt*-ABs harvested in the stationary phase, indicating the potential conservation DNA integrity during this stage.

3) Line 371, what's the meaning of NA?

We apologize to the reviewer. We used NA as acronym for Nucleic Acids, but as this term only is used very few times in the manuscript, we now changed “NA” to “nucleic acids”.

4) The authors mention the genes and proteins that regulate ABs production. Can they apply transcriptomic and proteomic analyses to study the functions of these genes and proteins in ABs?

We thank the reviewer for this comment. Indeed, we plan to perform proteomics studies on the proteins in *Gt*-ABs in future studies.

Reviewer #4 (Remarks to the Author):

Corredor and colleagues investigate whether apoptotic bodies are generated by aged Cryptophytes, specifically *Guillardia theta*, a unicellular microalga, using Confocal Microscopy, Transmission Electron Microscopy (TEM), Fluorescence-Activated Cell Sorting-FACS, and Confocal Raman Spectroscopy (RS). The authors have done an excellent job in identifying released bodies and its content, providing well-documented evidence that represents an important advance in the field of cellular communication in unicellular organisms.

Demonstrating the release of apoptotic bodies would be a novel finding. However, to determine whether these structures are truly apoptotic bodies or extracellular vesicles that facilitating cell-to-cell communication in this aged organism, like others found in another algae or cyanobacteria (Paterna, et al. 2022, DOI:10.3389/fbioe.2022.836747,

Biller et al. 2014, DOI:10.1126/science.124345), it is essential to follow the recommendations of the Nomenclature Committee on Cell Death (NCCD), cited in the manuscript, and, for example, those outlined by Aguilera et al., 2021,- DOI: 10.3389/fmicb.2021.631654 i.

We thank the reviewer for this comment. We now added a table (Supplementary Data Table 2) to the manuscript listing the various vesicles found in different organisms to show the similarity of Gt-ABs to metazoan ABs and their difference to other types of vesicles. We added to the manuscript the expression analysis of the metacaspases I and III (structural homologues to the metazoan caspases) (new Figure 1, panel C) we additionally performed further experiments, to show the pharmacological induction of programmed cell death using the well-known apoptosis inducers staurosporine and carbonyl cyanide m-chlorophenyl hydrazone, which also resulted in the generation of Gt-ABs (Extended Data Figure 2).

Some of my major concerns in this regard:

The study of apoptotic bodies of Corredor and colleagues is supported by morphological and size-based evidence. However, according to the NCCD guideline, biochemical, genetic, and pharmacological approaches should be integrated in order to robustly support the statement.

We agree with the reviewer and now added molecular and pharmacological approaches as indicated above. Depolarization of the mitochondrial membrane using the dyes JC-10 and TMRM was examined, as explained in detail in the following page. We also would like to point out that the study presented in the manuscript includes a thorough biochemical characterization on the composition of the Gt-ABs using Confocal RAMAN Spectroscopy. Our findings highlight the similarity of the contents of Gt-ABs and the reported composition of metazoan ABs.

- The exploration of molecular markers or signalling pathways associated with apoptosis in *G. theta* could further substantiate these findings. Including such analyses would provide more robust evidence supporting the apoptotic origin of the observed bodies.

We agree with the reviewer, unfortunately the metabolism of cryptophytes is still enigmatic. Cryptophytes like *G. theta* are microalgae derived after secondary endosymbiosis, rendering them to contain four genomes; the nucleus, mitochondria, chloroplast and the nucleomorph, a relic nucleus remnant of the red algae, which had been engulfed during the second endosymbiosis. As their name states, cryptophytes are a cryptic type of algae. To this day there is no exact branch of the eukaryotic tree of life, in which cryptophytes can be placed. To cite John M. Archibald: “On the basis of plastid-associated features, such as pigmentation and molecular sequence data, cryptomonads have long been linked to other chlorophyll c-containing algae such as haptophytes and plastid-bearing stramenopiles (e.g., golden/ brown algae and diatoms). But the latest phylogenomic analyses based on nuclear sequence data do not support this affiliation — cryptomonads (including the plastid-lacking goniomonads) instead appear to be more closely related to heterotrophic (and plastid-lacking) protists such as katablepharids. Clearly, cryptomonads diverged from all other known eukaryotes a long, long time ago, and how their red alga-derived plastids are related to those of other secondary plastid-bearing algae is still far from clear. The internal phylogeny of cryptomonads is also still very much a work in progress, which complicates efforts to study emerging phenomena” (10.1016/j.cub.2020.08.101). (An example to highlight their difference to other photosynthetic organisms and their evolutionary uniqueness: while the light

harvesting antenna in green plants is integrated into the chloroplast thylakoid membrane or in red algae is situated at the outside of the thylakoid membrane, in G. theta it is located within the thylakoid lumen.) Therefore, cryptophytes are very interesting organisms worth studying, but current data are not available.

We performed a bioinformatic study (data not shown) to find in the G. theta genome homologues of genes associated with PCD processes in metazoans and other organisms such as: Pannexin 1 (PANX1) channels, kinase (ROCK1), Caspase activated DNase_(CAD), Inhibitor caspase activated deoxyribonuclease (DNA fragmentation related from ICAD complex), among others. Unfortunately, although G. theta is a model organism, it's genome is still a work in progress and highly fragmented. We had hits for certain genes but with low statistics. At this time, we cannot say with confidence that the hits correspond to true gene homologues. In the future, we would like to perform an extremely thorough bioinformatic analysis to be able to explore the molecular markers or signalling pathways associated with apoptosis in G. theta and be able to report it with high confidence.

- The inclusion of pharmacological studies in the manuscript would significantly enhance the robustness of the findings. For instance, the use of apoptosis inducers such as staurosporine (STS) or caspase inhibitors could help elucidate whether caspases are involved in the formation of apoptotic bodies in G. theta. While most apoptosis inducers and inhibitors so far have been exclusively used on metazoans, some have been applied to the photosynthetic eukaryotic and prokaryotic organisms for demonstrating specificity of a particular type of cell death (Dixon et al 2012, 10.1016/j.cell.2012.03.042, Ramachandran et al 2023, DOI: 10.3390/cells12040553, Aguilera et al 2022, DOI: 10.1083/jcb.201911005, Carbó et al 2023, DOI: 10.1016/j.jhazmat.2023.130997)

We thank the reviewer for the comment. We induced apoptosis in healthy G. theta cells in the exponential growth phase using staurosporine (5 µM) or the protonophore carbonyl cyanide m-chlorophenyl hydrazone (49 µM) and examined their morphology and the generation of apoptotic bodies for 6 hours. The results of these experiments as well as timelapse images with TL-DIC microscopy of the apoptotic cells are in agreement with our observations at natural aging and have been added to the manuscript (Extended Data Figure 2). G. theta cells were also incubated with up to 80 µM of cycloheximide, which did not induce cell death and therefore was not analysed further. Regarding caspase inhibitors, G. theta does not encode caspases, but structural caspase-homologues, termed metacaspases. We tested the effect of the metacaspase inhibitor VRPR-FMK (<https://doi.org/10.1073/pnas.2303480120>). Healthy cells growing in the exponential phase, were centrifuged, washed and resuspended in artificial seawater, as described above (experiment describing addition of Gt-ABs to G. theta cells in the exponential phase). Cultures with a density of 1x10⁶ cells/ml were grown for 16 days in the presence of 1 µM of Z-VRPR-FMK (Enzo Life Sciences). A culture with DMSO (vehicle) was used as a negative control. Formation of Gt-ABs was counted with a Multisizer 3 Coulter counter (Beckman Coulter). Although we measured increased metacaspase expression, we did not observe any difference in the culture after addition of the metacaspase inhibitor (Figure C), neither on cell death nor Gt-ABs production. Literature data on in vivo experiments using Z-VRPR-FMK is limited, it therefore might be possible that the compound did not enter the cells.

Figure C: Addition of the metacaspase inhibitor Z-VRPR-FMK to a living *G. theta* culture (in vivo).

Line 318. Some assays to measure membrane potential and organelle functionality could be highly interesting and valuable for this study (eg., JC-1 staining)

We thank the reviewer for this comment. We used two specific dyes to detect mitochondria membrane depolarization in algal cells; JC-10, a JC-1 derivative (Enzo Life Sciences) with improved solubility, reduced non-specific aggregation and enhanced reproducibility and tetramethylrhodamine (TMRM), which relies on a distinct, intensity-based detection strategy. JC-10 is excited at 510 nm, with fluorescence detected at 525 nm (green monomer) and 590 nm (orange aggregate). The excitation/emission spectra for TMRM are 548/574 nm. The assay compared *G. theta* cells in the exponential growth phase (healthy) and aged cells, each with and without treatment with 25 μ M carbonyl cyanide 3-chlorophenylhydrazone (CCCP), used as a positive control for mitochondrial membrane depolarization.

Method: For sample preparation, a modified protocol from <https://doi.org/10.1007/s10811-019-01860-3> was used for the microplate assay, confocal microscopy, fluorescence lifetime imaging microscopy (FLIM) and FACS. The cells were centrifuged at 7000 rcf for 3 minutes at room temperature, washed and resuspended in PBS buffer, pH 7.4, in a 1×10^6 cells/mL concentration. Mitochondria were also stained with 100 nM TMRM. *G. theta* cells (5×10^5 cells/mL) in exponential and death phase were resuspended in PBS, and incubated with TMRM for 30 minutes at room temperature. For the microplate assay a Corning Costar® 96-Well Black Polystyrene Plate was used and fluorescence was monitored in a BioTek Synergy™ H4 microplate reader.

Fluorescence measurements showed that aged cells had a lower orange-to-green ratio compared to exponential cells (Figure D), indicating depolarized mitochondrial membrane and a shift from aggregated (orange) to monomeric (green) JC-10 fluorescence. However, CCCP-treated cells displayed even higher fluorescence levels. Similar results were obtained with TMRM staining. The pronounced fluorescence in CCCP-treated samples may result not only from mitochondrial ATPase uncoupling, but also from CCCP's inhibitory effect on electron flow in Photosystem II (e.g. [https://doi.org/10.1016/0014-5793\(94\)01309-Q](https://doi.org/10.1016/0014-5793(94)01309-Q)), leading to excess light energy being re-emitted as fluorescence. Confocal imaging of the JC-10- and TMRM-stained cells supported the microplate assay results.

Figure D: Trial measurements of the mitochondrial functionality were not possible due to chloroplast fluorescence.

Furthermore, FACS analysis was performed on algal cells during the exponential and death phases, with and without CCCP treatment and on both JC-10-stained and unstained samples. The cell suspension was analyzed for green monomers (excited at 488 nm; emission detected using a 530/30 filter with a 502 nm long-pass dichroic mirror) and orange aggregates (excited at 561 nm; emission detected using a 582/15 filter). As displayed in the Figure E, CCCP treatment caused an increase in red and green fluorescence-positive cells in both healthy and aged cell cultures. The addition of JC-10 dye to CCCP-treated exponential cells resulted in green-only and red-only fluorescent populations, while the amount of the green-and-red population slightly decreased. JC-10-stained cells in exponential growth consisted of green (8.2%) and red-and-green (37.9%) populations, while in JC-10-stained cultures of the death phase the most prominent population was the green-positive (57.4%), with the red-and-green population accounting for 23.1 % of the total events. The rise of green fluorescence signal in aged cells, compared to healthy cells, coincided with the accumulation of green monomers in the depolarized apoptotic mitochondria.

Figure E: FACS analysis on JC-10 stained cells

In order to investigate the interference of chlorophyll autofluorescence in CCCP-treated and JC-10 stained cells we used FLIM microscopy (STELLARIS 8 DIVE, Leica Microsystems). Separation of the fluorescent species using the TauSense tool was inconclusive, since the lifetime of JC-10 is unknown.

In conclusion, we observed a collapse of the mitochondrial membrane potential in aged G. theta cells, in agreement with early apoptosis. However, as our positive control (CCCP-treated cells), with the methods and the conditions tested, failed to yield the expected results the validity of the results cannot be

ensured. Spectral overlap with chlorophyll autofluorescence induced by the inhibitory effect of CCCP compromise the results.

Line 324-343. I found the paragraph between lines 324-343 somewhat contradictory. Firstly the authors list some important differences that exist between apoptotic cells in metazoans and in *G. theta*, but the conclusion state that "*G. theta*'s apoptotic morpho-phenotype is highly identical to metazoan apoptosis". Please clarify.

We apologize for the lack of clarity. Originally, in the paragraph above the one mentioned by the reviewer, we listed all similarities between metazoan ABs and Gt-ABs. In this paragraph, we then stated that single-cell apoptosis is more simple than metazoan apoptosis, give examples and give the conclusion of both paragraphs. We now rephrased and restructured the text and hope it is easier to understand.

Christiane Funk, Ph.D.
Professor in Biochemistry

Point-by-point response to the reviewers' comments

Please show all changes in the manuscript text file with track changes or colour highlighting. If you are unable to address specific reviewer requests or find any points invalid, please explain why in the point-by-point response.

First of all, we would like to thank all reviewers for their time and effort and for their positive feedback. The concerns of Reviewer 1 are addressed below.

REVIEWER COMMENTS

Reviewer #1 (Remarks to the Author):

I am generally satisfied with the new version of the manuscript, as the authors took good care at addressing most of the points that we have raised.

I still have some reserves and important points related to the EM work:

- major point: sorry to insist, but neither from the text, nor from the figure legend, it appears clear to me that the ABs are imaged from FACS sorted small particles or from whole cultures that include cells. If the latter, then I am afraid it is not possible to unambiguously identify free floating AB. A 2D plane in TEM can section and display a budding Abs without the showing cell it's budding from.

In the legend, authors write "Extracellular Gt-ABs, free floating in solution, .." and in the text "Extracellular, free-floating Gt-Abs ..." such sentences do not clearly state if those originate from a sorting of particles that would exclude the whole cells.

From these I would tend to interpret that EM was done only on cell cultures and not on sorted / filtered ABs.

Yet, looking at the methods section, it seems that TEM experiments were performed on both whole cultures and on purified ABs. The results should clearly state which conditions the images are coming from.

We thank the reviewer for these comments, and we welcome the opportunity to communicate our results as clearly as possible to maximize the impact of our research and further improve the quality of our manuscript.

In the results section, “*G. theta* cells display the ultrastructural markers of apoptosis and AB production”, we have included a phrase to explain that we used both types of samples, whole cultures and filtered, enriched cultures with Gt-ABs (as mentioned in the methods section), to track the ultrastructural changes in the cell population during aging and to analyse the detailed ultrastructure of the ABs. In addition, in lines 135-136, we now state that extracellular, free-floating Gt-ABs originate from filtered, enriched cultures. This information we also added to the legend of Figure 2.

The aim of this manuscript section was to display the ultrastructural changes of *G. theta* cells during the apoptotic process, as well as the production of Gt-ABs. The use of only filtered and sorted Gt-ABs would have prevented us from observing those ultrastructural changes, however, we would like to emphasize that at the last time point (day 42), after filtration, the samples constituted predominantly Gt-ABs.

We do agree with the reviewer that a 2D plane in TEM can potentially section and display biases. For that reason in the Extended Data Figure 1 we have included images taken with different microscopy techniques. For further clarity, we have specified in the legend of the Extended Data Figure 1 that all the images in the column to the right (panels B, D, F, H) correspond to sorted populations of ABs. The following TEM image also corresponds to sorted Gt-ABs, as an additional example.

Figure A: Transmission electron microscopy of sorted Gt-ABs.

In addition, we would like to refer to lines 77-81 of the results section “Apoptosis and the production of apoptotic bodies (ABs) in *G. theta*”:

“During experiments with *G. theta* cultures grown in standard conditions, we observed the presence of extracellular vesicles in solution (the extracellular space). In stationary and death phases of the culture, when the cells experienced nutrient depletion and ageing, the proportion of these vesicles (hereafter *G. theta* Apoptotic Bodies (Gt-ABs)) increased and was accompanied by the progressive decrease in cell number”.

For further confirmation the following images (transmitted light and confocal microscopy) show *G. theta* cells and Gt-ABs in solution (free-floating in the

extracellular space). Due to space limitations we did not add these photos to the manuscript.

Figure B: Transmitted light microscopy of free-floating *G. theta* cells and Gt-ABs present in solution.

Figure C: Fluorescent confocal microscopy of free-floating *G. theta* cells and Gt-ABs present in solution. *G. theta* cells display chlorophyll autofluorescence in magenta. Gt-ABs show absence of pigments.

As mentioned in our previous response, this study (neither filtering nor sorting) would not have been possible if the Gt-ABs were attached to the cells. To emphasize this fact, we now have added a short sentence to the results section “Classification and characterization of Gt-ABs by FACS” explicitly stating that “The Gt-ABs were sorted in a single-particle format” (Line 181). This method does not allow sorting of clumps or aggregated particles. As shown in Figure D, post-sorting only Gt-ABs were observed.

Figure D: Transmitted light microscopy of sorted Gt-ABs.

Therefore our statement “Extracellular, free-floating Gt-ABs” is based on evidence confirmed by various techniques:

- 1. Transmitted Light Microscopy**
- 2. Fluorescence Microscopy**
- 3. Fluorescence Confocal Microscopy**
- 4. Transmission Electron Microscopy**
- 5. Confocal RAMAN Spectroscopy**
- 6. Fluorescence-Activated Cell Sorting (FACS)**
- 7. Multisizer 3 Coulter counter (Beckman Coulter)**

- minor point: on figure 2, please be consistent in labelling the nuclei or nucleoli. "N" is used in the first panels and this not defined in the legend. I assume they refer to the nuclei there.

We apologize and thank the reviewer for pointing out this mistake. During the production of images with labels that are more legible to the reader (as suggested in the previous revision) part of the acronyms were cut unintentionally. We now use the following nomenclature: N — nucleus, Ns — nuclear space, No — nucleolus, NoS — nucleolus space (see legend Figure 2).

The labels in Figure 2 have been corrected, and the figure has been replaced in the main manuscript document. The corrected high-resolution figure will also be uploaded to the system at the time of resubmission, to ensure good visualization of the corrected details.

In addition, to keep consistency with the labels, also Extended Data Figure 1 has been changed accordingly.